



# Ammonia measurements from space with the Cross-track Infrared Sounder (CrIS): characteristics and applications

Mark W. Shephard[1], Enrico Dammers[1], Karen E. Cady-Pereira[2], Shailesh K. Kharol[1,3], Jesse Thompson[1], Yonatan Gainariu-Matz[1,4], Junhua Zhang[1], Chris A. McLinden[1], Andrew Kovachik[1,4], Michael Moran[1], Shabtai Bittman[5], Christopher Sioris[1], Debora Griffin[1], Matthew J. Alvarado[2], Chantelle Lonsdale[2], Verica Savic-Jovcic[1], and Qiong Zheng[1].

[1]Environment and Climate Change Canada, Toronto, Ontario M3H 5T4, Canada.
[2]Atmospheric and Environmental Research (AER), Lexington, MA, USA.
[3]University of Toronto, Toronto, Ontario, Canada
[4]University of Waterloo, Waterloo, Ontario, Canada.
[5]Agriculture and Agri-Food Canada (AAFC), Agassiz, British Columbia, Canada

*Correspondence to*: Mark W. Shephard (Mark.Shephard@canada.ca)

**Abstract.**

Despite its clear importance, the monitoring of atmospheric ammonia, including its sources, sinks and links to the greater nitrogen cycle, remains limited. Satellite data are helping to fill the gap in monitoring from sporadic conventional ground and aircraft-based observations, to better inform policymakers, and assess the impact of any ammonia-related policies. Presented is a description and survey that demonstrate the capabilities of the CrIS ammonia product for monitoring, air quality forecast model evaluation, dry deposition estimates, and emissions estimates from an agricultural hotspot. For model evaluation, while there is a general agreement in the spatial allocation of known major agricultural ammonia hotspots across North America some high-latitude regions during peak forest fire activity often have ammonia concentrations approaching those in agricultural hotspots. The CrIS annual ammonia dry deposition in Canada (excluding Territories) and the U.S. have average and annual variability values of $\sim 0.8 \pm 0.08$ Tg N year$^{-1}$ and $\sim 1.23 \pm 0.09$ Tg N year$^{-1}$, respectively. These satellite derived dry depositions of reactive nitrogen from $NH_3$ with $NO_2$ show an annual ratio of $NH_3$ compared to their sum ($NH_3 + NO_2$) of $\sim 82\%$ and $\sim 55\%$ in Canada and U.S., respectively. Furthermore, we show the use of CrIS satellite observations to estimated annual and seasonal emissions near Lethbridge, AB, Canada a region dominated by high emission feedlots also referred to as Concentrated Animal Feeding Operations (CAFOs); the satellite annual emission estimate of $37.1 \pm 6.3$ kt/yr is at least double the value reported in current bottom-up emission inventories for this region.

# 1 Introduction



Ammonia (NH₃) is the most abundant alkaline gas in the atmosphere and has major impacts on air, soil, and water quality. Ammonia generally reacts quickly with available acids (e.g. nitric and sulphuric acid from NO$_x$ and SO$_x$) to produce a significant fraction (~50%) of the secondary fine particulate matter (diameter < 2.5 µm) (e.g. Seinfeld and Pandis, 1988), which has significant human health impacts (e.g. cardiovascular and respiratory diseases) (Schwartz et al., 2002; Reiss et al., 2007; Pope et al., 2000, 2002, 2009; Crouse et al., 2012). In addition to air quality impacts, NH₃ and the aerosols formed from it (e.g. ammonium (NH4$^+$), ammonium nitrate NH₄NO₃ and ammonium sulphate (NH₄)₂SO₄) deposit from the atmosphere to the surface. Deposition of reactive nitrogen (N$_r$) into ecosystems provides important nutrients (e.g. increases crop production) and helps feed the world population, but even in small amounts it can have negative impacts on sensitive ecosystems, such as soil acidification (Galloway et al., 2003), eutrophication (Bergstrom et al., 2006), changes to vegetation type, and biodiversity loss (Fenn et al., 2010; Bowman et al., 2012; Sheppard et al. 2011; Bauer et al., 2016). Ammonia gas is phytotoxic to certain plant species, mainly bryophytes and lichens typical of nutrient poor natural ecosystems such as alpine areas and boggs.

Despite the importance of ammonia, historically anthropogenic emissions of NH₃ have largely been unregulated, especially outside of Western Europe, which has contributed to the lack of observations and the associated large uncertainties in our knowledge of ammonia fluxes. Furthermore, traditional in-situ surface-based monitoring of atmospheric ammonia is challenging due to the sticky nature of the ammonia molecule and the labour intensive and high-cost of performing the measurements, which result in making ammonia observations being sparse over most regions, especially in remote locations. Combining this relative scarcity of observation networks with the typically high spatial and temporal variations of atmospheric ammonia concentrations, owing to its short lifetime (hours to a day) and numerous diffuse agricultural sources, leads to an overall lack of knowledge of ammonia compared to other common air quality contaminants with the consequence that ammonia and ammonium are among the least known parts of the ecosystem's nitrogen cycle (Erisman et al., 2007).

Recent satellite observations of lower tropospheric ammonia are helping fill in observational and knowledge gaps. Satellite observations of lower tropospheric ammonia has only been possible in the past decade through concurrent improvements in both the radiometric capabilities of infrared instruments on satellites and the radiative transfer forward modelling and inversion algorithms. This was first demonstrated by Beer et. al., (2008) with NASA's Tropospheric Emission Spectrometer (TES) observations, which has since been followed by ammonia observations from the European Space Agency (ESA) Infrared Atmospheric Sounder Interferometer (IASI) (Clarisse et al., 2010), the NASA/NOAA Cross-track Infrared Sounder (CrIS) (Shephard and Cady-Pereira, 2015),the NASA Atmospheric Infrared Sounder (AIRS) (Warner et al., 2016), and JAXA Greenhouse Gases Observing Satellite (GOSAT) (Someya et al., 2019). Out of this suite of instruments, AIRS, CrIS, and IASI all have large swaths providing daily global coverage. Furthermore, both IASI and CrIS are part of operational meteorological platforms with re-flights of instruments on several satellites in succession enabling multi-decadal time series (e.g. planned coverage spanning: 2006-2021 for IASI; 2011-2038 for CrIS, and 2021-2042 for the next generation IASI-NG). CrIS is the newest of these operational satellites and has the lowest radiometric noise in the portion of the spectrum used for ammonia retrievals (Zavyalov et al., 2013). This provides CrIS with the potential for increased vertical sensitivity of ammonia





near the surface along with global coverage. Presented here is a current survey of the CrIS $NH_3$ Fast Physical Retrieval (CFPR) product characteristics with some example applications.

## 2 CrIS satellite retrievals

The CrIS instrument is a Fourier Transform Spectrometer (FTS) launched by the U.S. NOAA/NASA on both the Suomi National Polar-orbiting Partnership (S-NPP) satellite on 28 October 2011, on the NOAA-20 satellite on 29 November 2017. Here we focus only on the longer time series of data provided by the CrIS instrument flown on S-NPP. The S-NPP satellite is in a sun-synchronous low earth orbit with overpass times of ~1:30 and 13:30 mean local time. CrIS is a hyperspectral infrared instrument with a spectral resolution of 0.625 cm$^{-1}$. The main advantage of CrIS is the combination of dense global coverage and the improved sensitivity in the boundary layer due to the low spectral noise of ~0.04 K at 280K in the $NH_3$ spectral region (Zavyalov et al., 2013) and the early afternoon overpass which coincides with high thermal contrast (difference between the surface and air temperature) when the infrared instrument is more sensitive. A detailed description of the CrIS Fast Physical Retrieval (CFPR) algorithm for deriving ammonia applied to both simulated spectra and initial sample observations was provided by Shephard and Cady-Pereira (2015). Since then the CFPR algorithm has been applied globally to CrIS spectra from May 2012 onwards. The input atmospheric state required for the radiative transfer forward model calculations are obtained from the Level 2 Cross-Track Infrared and Microwave Sounding Suite (CrIMSS) Atmospheric Vertical Profile Environmental Data Record (product ID: REDRO) (Divakarla et al., 2014) product from May 1, 2012 to April 7, 2014, after that the retrieved Level 2 NESDIS-unique CrIS-ATMS product system (NUCAPS) (Liu et al., 2014) are used. The CPFR retrieves the surface temperature and emissivity for each observation (field-of-view) prior to the ammonia retrieval. Ammonia profiles are retrieved at 14 profile levels to capture the vertical sensitivity of ammonia that varying from profile-to-profile depending on the atmospheric conditions. The CrIS satellite ammonia observations do not have equal sensitivity in the vertical, and have coarse vertical resolution (e.g. ~1 to 3-km). Hence, surface level values and total column values are both highly correlated with the boundary layer values where the satellite typically has peak vertical sensitivity. Note that atmospheric ammonia is typically short-lived so that higher concentrations are generally close to the sources, which are generally near the surface. This is demonstrated later in Section 3.2 with model emissions and corresponding simulated surface concentrations.

An update from the initial Shephard and Cady-Pereira (2015) analysis is that under favourable conditions CrIS detects $NH_3$ near surface concentrations down to ~0.3-0.5 ppbv (e.g. Kharol et al., 2018), which is less than half of the more conservative estimate of ~1 ppbv previously reported using an Observation System Simulation Experiment (OSSE). This is mainly due to the better than specified noise capabilities in the observed CrIS $NH_3$ spectra, and the limited number of sampling conditions used in the original OSSE experiment.

Since he CFPR uses a mathematically robust physics-based optimal estimation framework (Rodgers, 2000) it provides the vertical sensitivity and the measurement information content (obtained from the averaging kernels), and an estimate of the





retrieval errors (error covariance matrices), for each observation. The output sensitivity and error parameter characterization are key for utilizing CrIS observations in air quality model applications such as data assimilation, data fusion, and model based emission inversions (e.g. Li et al., 2019). It is also important that, as done first in the TES NH$_3$ retrieval (Shephard et al., 2011), the CFPR algorithm uses only three *a priori* ammonia profiles. These *a priori* profiles represent unpolluted, moderate,

5   and polluted conditions with no prescribed latitudinal or seasonal dependence. For each retrieval one of these three *a priori* profiles is selected based on the estimated ammonia spectral signal, as there is little known about ammonia globally (i.e. there is no spatial climatology field used for the *a priori* as is commonly done for retrievals of better known species). The retrieval quality flags are described in Appendix A.

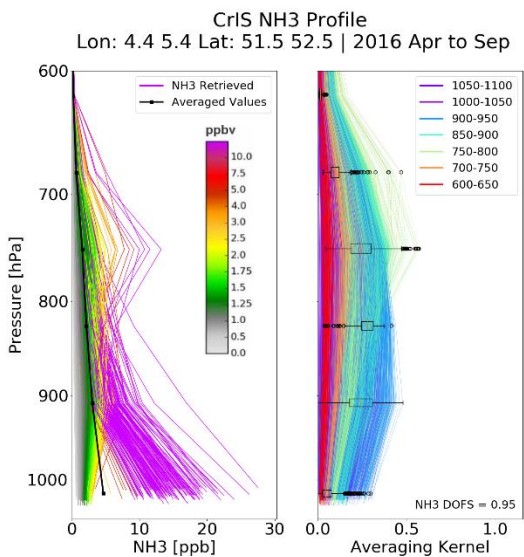

Figure 1. CrIS retrieved NH$_3$ profiles from a 50km radius around Cabauw, Netherlands from April-September 2016. The retrieved profile level values below 600 hPa are shown in the left panel and coloured according to the surface value, and the corresponding rows of the averaging kernels are shown in the right panel.

15       As shown in Figure 1 the peak sensitivity is generally in the boundary layer below ~700 hPa (3-km). The lower instrument noise with similar spectral resolution offered by CrIS allows for greater sensitivity near the surface with less dependency on the thermal contrast from an operational meteorological sensor. This also follows from simulation studies performed by Clarisse et al. (2010) that show with even twice the CrIS noise level there would be a significant reduction in the dependence on thermal contrast for sensitivity in the daytime boundary layer. As there is generally only ~1 degree-of-

20   freedom for signal (DOFS) (e.g. 0.95 average in Figure 1) with coarse vertical resolution (half-width-at-half-maximum of the rows of the averaging kernels) of ~1to 3-km, the retrieved surface level concentrations are highly correlated with the retrieved levels at higher elevationsin the boundary layer. The retrieved profiles in Figure 1 still tend to have distributions that are



grouped around the three *a priori* profiles and so, future updates to the retrieval will investigate various refinements to the *a priori* profiles and constraints. Figure 2 shows a sample single day scene of NH$_3$ retrievals on May 15, 2016 during the Fort McMurray fires (Adams et al., 2019) ranging from low background values of < 1 ppbv to elevated values up to 30 ppbv. Corresponding Cloud-Aerosol Lidar and Infrared Pathfinder Satellite Observation (CALIPSO) (Winker et al., 2003) lidar

measurements on this day shows the smoke plume reaching altitudes above the ground of ~ 2.0 to 3.0 km (~800 to 700 hPa) (for reference see Figure A 1 obtained from https://www-calipso.larc.nasa.gov/data/BROWSE/production/V4-10/2016-05-15/2016-05-15_19-42-56_V4.10_3_6.png).

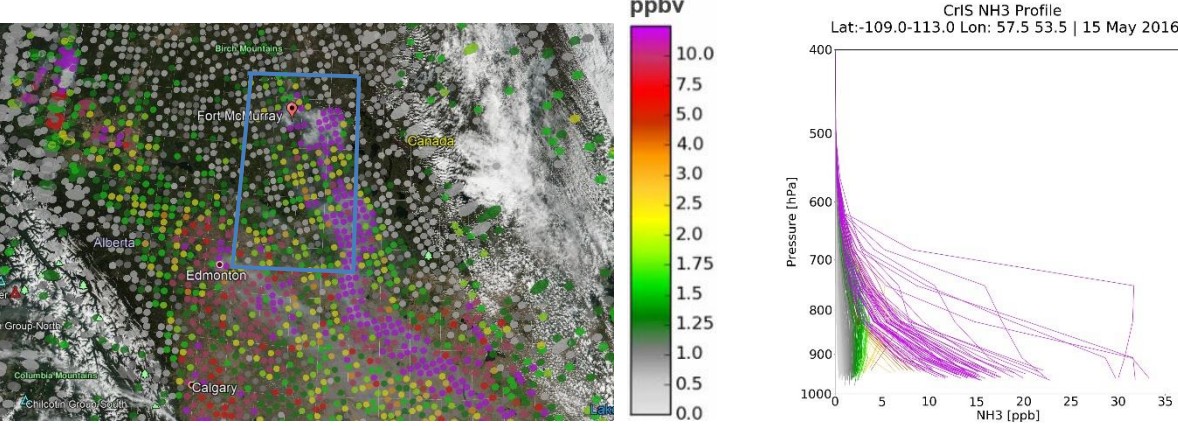

Figure 2. Overlay of the CrIS retrieved surface NH$_3$ pixels on the VIIRS true colour visible imagery for May 15, 2016 showing
the plume from the Fort McMurray fires (left panel). Retrieved NH$_3$ profiles colour coded using the surface level values in the blue box on the map (right panel). (Underlying VIIRS images obtained from NASA Worldview (https://worldview.earthdata.nasa.gov/))

The estimated random errors for both the observation (consisting of only measurement errors here as no cross-state
errors are estimated) and the total error (includes the measurement and representative (or smoothing) error are computed for both the individual retrieval profile levels and the integrated total column (see Shephard and Cady-Pereira, 2015). Observation errors can be used if the vertical resolution of the satellite observations are already taken into consideration (e.g. satellite observation operator is applied to the comparison dataset), whereas the total error should be used if the satellite retrieved value is to represent the discrete observation resolution (e.g. individual profile level value, or vertical column value were the vertical
sensitivity is not considered). Errors from a single day of global retrievals as a function of concentration amounts are provided in Figure 3 for total column, and in Figure 4 for values from each profile level in the boundary layer below 700 hPa (~3-km). For the total column amounts, the measurement errors are typically in the 10 to 15% range, whereas the total errors are ~30%. The individual profile retrieval levels have measurement errors of ~10%, except for low concentrations with amounts < 1 ppbv where the error rises to ~30%. When the smoothing error component is included for the individual profile levels, the profile





level total random error increase to the range of 60% to 100%. This is expected given that current ammonia nadir infrared retrievals have limited vertical information (resolution), leading to significant smoothing of the retrieved profile.

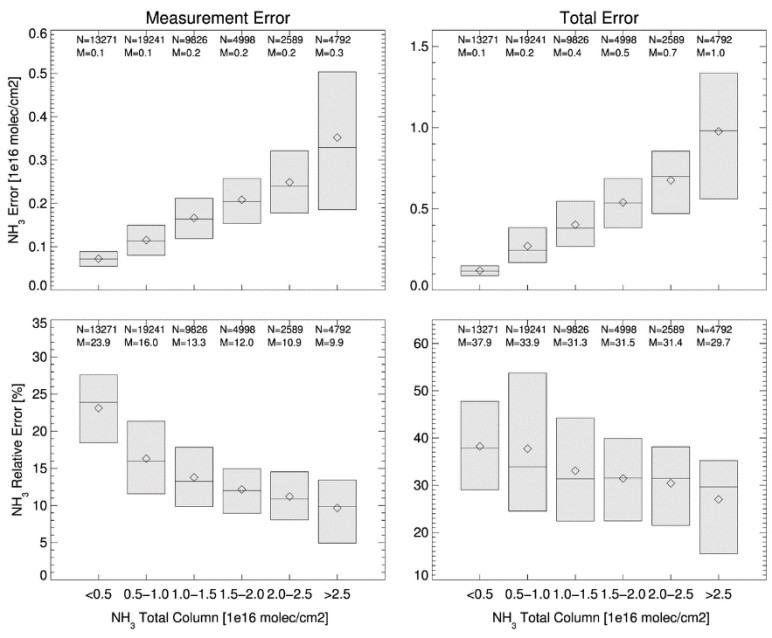

Figure 3. Estimated CrIS retrieved $NH_3$ total column measurement and total errors statistics from global values for May 28, 2017. Both absolute error estimates (top panels) and corresponding relative (fractional) errors (bottom panels) are shown. The diamonds are the mean values for each box range. Only retrievals over land with a quality flag of 5 (therefore DOFS $\geq 0.1$) where included. The N and M are the number of points and the median value for each box, respectively.





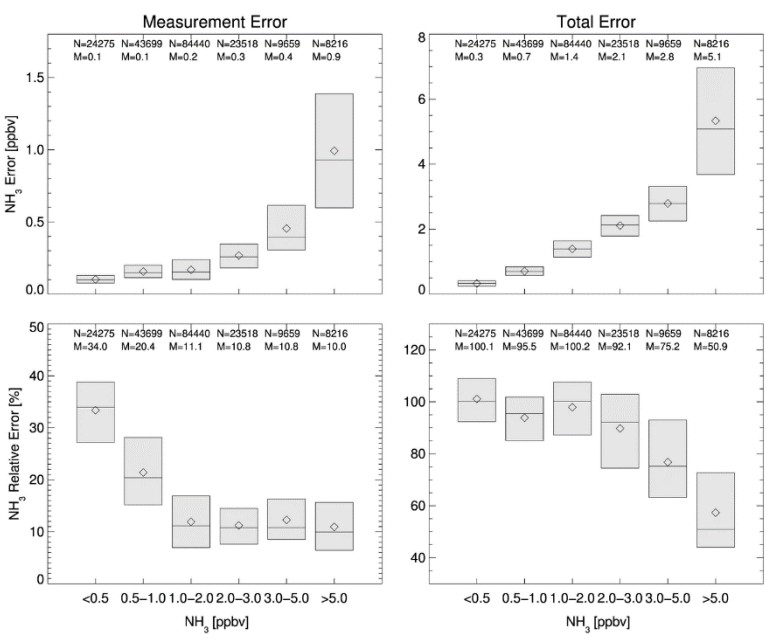

Figure 4. Errors for the individual for all profile levels below 700 hPa (~3-km) using the same plotting criteria as in Figure 3.

The CFPR (Version 1.3) was also validated by Dammers et al., (2017) against ground-based FTIR observations to determine the actual errors (as opposed to the estimated retrieval errors). These initial results show a good overall comparison results with a correlation of r~0.8 with a slope of 1.02. For retrievals with total column values > 1.0 x $10^{16}$ molecules cm$^{-2}$ (ranging from moderate to high levels) the relative bias difference is < 5%, with a standard deviation ranging from 25 to 50%. For total column comparisons for smaller values < 1.0 x $10^{16}$ molecules cm$^{-2}$ there are larger differences with a CrIS higher than the FTIR by ~30% with a standard deviation of ~40%. Initial CrIS comparisons of the surface level retrievals with in-situ surface observations from the Ammonia Monitoring Network (AMoN) over North America show a correlation of 0.76 and an overall mean CrIS – AMoN difference of ~+15% (Kharol et al., 2018). An extension of this analysis over more surface locations and longer time-periods are currently being performed. These CrIS retrieved ammonia profiles (Level 2 products) are used to generate gridded averaged (Level 3) products over various spatial grids and time-periods (e.g. monthly average on 0.1 x 0.1 degree latitude and longitude grid). To reduce discontinuities between adjacent grid points and increase the effective resolution of observations that are averaged over extended time-periods (e.g. monthly), oversampling with weighting (e.g. Gaussian weighting based on distance from the centre of the grid) (e.g. Fioletov et al., 2011; Pommier et al., 2013) is used in the generation of the Level 3 gridded averaged products. The output Level 3 product grid size and level of oversampling are flexible and based on the purpose and number of observations (e.g monthly, annual, multi-year). Figure 5 is a sample global plot of the CrIS gridded surface level NH$_3$ for the 2013-2017 time-period showing global "hot-spots" over land. Figure 5



shows the typical elevated large area NH₃ "hotspots" with annual values averaging over ~8 ppbv. These regions with high NH₃ concentrations include: the Indo-Gangetic Plains in India/Pakistan, the Nile Delta in Egypt, California's Central Valley and central U.S and Canada, the Comarca Lagunera and Los Altos de Jalisco regions in Mexico, and north central Colombia and the west coast of Peru in South America, the Po Valley in Italy and Ebro Valley in Spain, the Fergana Valley in Central Asia, the Mekong Delta region in southern Vietnam, south central Thailand, Indonesia, and regions in eastern China. CrIS sensitivity is also demonstrated through its capability to observe ammonia over agricultural regions in the southern part of the Australian continent and forest fire prone regions in the north tips of the Northern Territory and Queensland. Elevated ammonia amounts seen over some desert regions such as the Taklimakan Desert in China and Sahara Desert with no obvious local ammonia sources need further investigation.

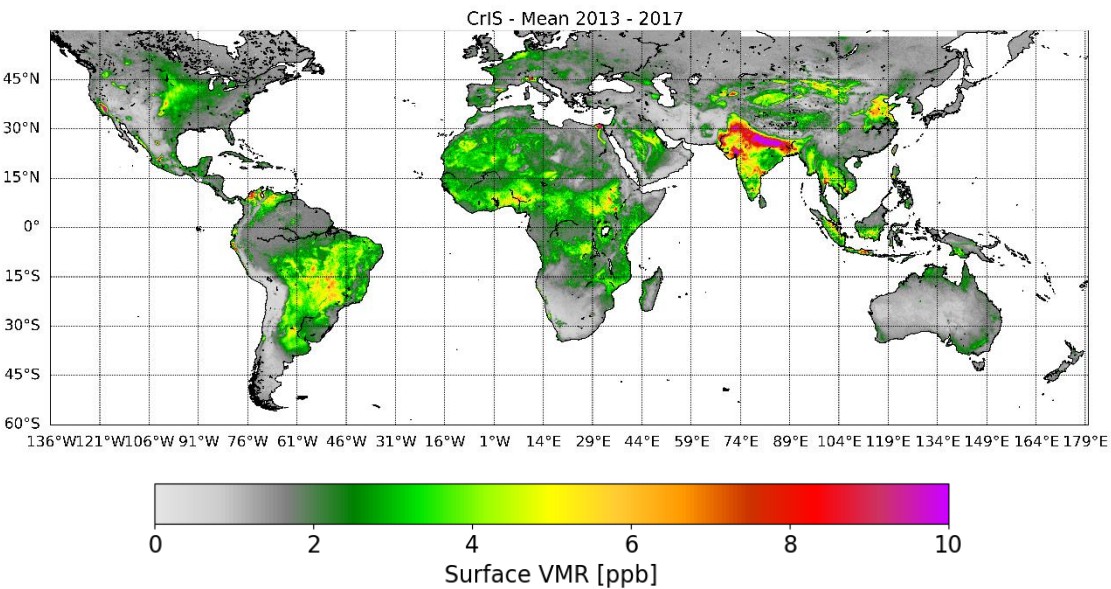

Figure 5. CrIS 5-year mean (2013-2017) of surface ammonia globally. The CrIS mean gridded Level 3 values are generated on uniform 0.05 x 0.05 degree (~5 x 5 km²) grid with a quality flag of 5.



## 3 Application examples

The CrIS NH$_3$ product can be used for many applications such as monitoring, air quality model evaluation (e.g. Whaley et al., 2018; Pleim et al., 2019), dry deposition estimates (Kharol et al., 2018), and emissions estimates for larger agriculture sources, industrial point sources (Dammers et al., 2018), and wildfires (Adams et al., 2019). Here we provide examples that

demonstrate and expand upon these applications.

### 3.1 Monitoring

As previously noted, the operational polar-orbiting satellites (e.g. IASI and CrIS) have the benefit of providing daily global spatial coverage on local to regional (e.g. 10's of km) scales over many decades that can help fill in gaps in current monitoring networks. Provided here are examples of daily, seasonal, and annual observations of ammonia by the CrIS satellite. While

not currently done so, it is possible to derive global daily ammonia products in near real time.

### 3.1.1 Daily

Ammonia in general is relatively short-lived in the boundary layer so its day-to-day atmospheric concentration levels over a region can vary greatly depending mainly on the meteorology (e.g. windspeed, temperature) and episodic events (e.g. biomass burning, spreading of fertilizer) and concentration of reactant acid gases. However, ammonia in the free-troposphere above

the boundary layer, for example during forest fires, is not quickly scavenged or deposited and hence has a longer lifetime and travels over large distances (Lutsch et al., 2019). In addition, ammonia deposited on certain surfaces (e.g. vegetation) can be re-released into the atmosphere later depending on the ammonia balance between the air and leaf apoplastic concentrations (compensation point) (do that there is bi-directional flow) (e.g. Massad et al., 2010; Bash et al., 2013; Pleim et al., 2019). Figure 6 shows CrIS surface NH$_3$ over North America for four consecutive days from July 5-8, 2014 and the corresponding

Aqua Moderate Resolution Imaging Spectroradiometer (MODIS) true-colour visible imagery with thermal hotspots (e.g. forest fires) overlaid in red. The elevated ammonia concentrations released from forest fires identified from the thermal hot spots and smoke plumes in the MODIS imagery are in the CrIS maps for north central Canada on all four days. The impact of wind on the location of the forest fire plumes is also visible. The daily maps also show the impact of cloud cover on the CrIS NH$_3$ retrievals, since optically thick clouds block the weaker ammonia signal from below the clouds.

25        In Version 1.5 of the CFPR NH$_3$ retrievals the observations with insufficient ammonia signal in the measured spectrum, mainly due to ammonia concentrations below the sensor detection limit (< 0.3-1.0 ppbv) or clouds blocking the ammonia signal, are not currently being processed. Thus, cloud filtering is presently achieved implicitly through the threshold of the ammonia signal in the spectra (i.e. no ammonia spectral signal through clouds) and the use of a surface brightness temperature threshold derived from global seasonal climatological cloud top temperatures (based on International Satellite

Cloud Climatology Project (SCCP) maps; https://isccp.giss.nasa.gov/products/browsed2.html). This upfront cloud screening also improves the data processing rate. Comparing the MODIS true-colour imagery with the CrIS NH$_3$ observations in Figure



6 demonstrates that this technique is very effective for cloud screening. Note that thin clouds (cloud optical depth < 1.0) that are near the surface with cloud-top temperatures close to the surface temperature still impacts the current ammonia retrievals, but in general has a non-significant impact on the overall results as seen in the examples in Figure 6. Algorithm refinements such as directly incorporating a newly developed coincident VIIRS cloud products to distinguish the pixels with no ammonia

5    signal due to cloud from those that have concentrations levels below the detection limits are presently being tested. For regions with low concentrations, this has the potential to increase the density of observations included while tending to decrease mean background values.







Figure 6. Single daily maps of MODIS true colour visible imagery and corresponding CrIS surface NH₃ observation for four consecutive days from June 1-4, 2016. Overlaided on the MODIS visible images are the red thermal hotspots indicating forest fire locations. (MODIS images obtained from NASA Worldview (https://worldview.earthdata.nasa.gov/)).



### 3.1.2 Seasonality

Ammonia concentrations in the atmosphere are influenced by agricultural practices and meteorological conditions. Ammonia emissions differ over the course of the growing season due to changing farming practices and ambient temperature, leading to a month-to-month variation in concentrations. As an example, the relative spatial seasonal variability over North

5 America is shown in Figure 7. For most of North America, there is often an increase in concentrations during the springtime associated with fertilizer and manure applications, and warming surfaces, at the start of the growing season, which shifts from April in the southern to central parts of the U.S. to May in the norther states and most of Canada. There can also be an increase over some source regions (i.e. U.S. Midwest, Idaho, Washington state) in the summertime associated with increased temperatures and certain farm practices like cleaning corals and manure storages and spreading manure on harvested winter

10 crops or forages in mid to late-summer promoting more volatilization. Also apparent in the plots is the increase in concentrations in the non-agricultural northern latitude regions during the drier summer season associated with wildfires, which can inject ammonia with minimal acid reactants into the free troposphere allowing the transport of ammonia over larger regions (Lutsch et al., 2019). The retrieved elevated concentration values at high-elevation over the Rocky Mountains in the wintertime needs to be further investigated as a potential retrieval issue.





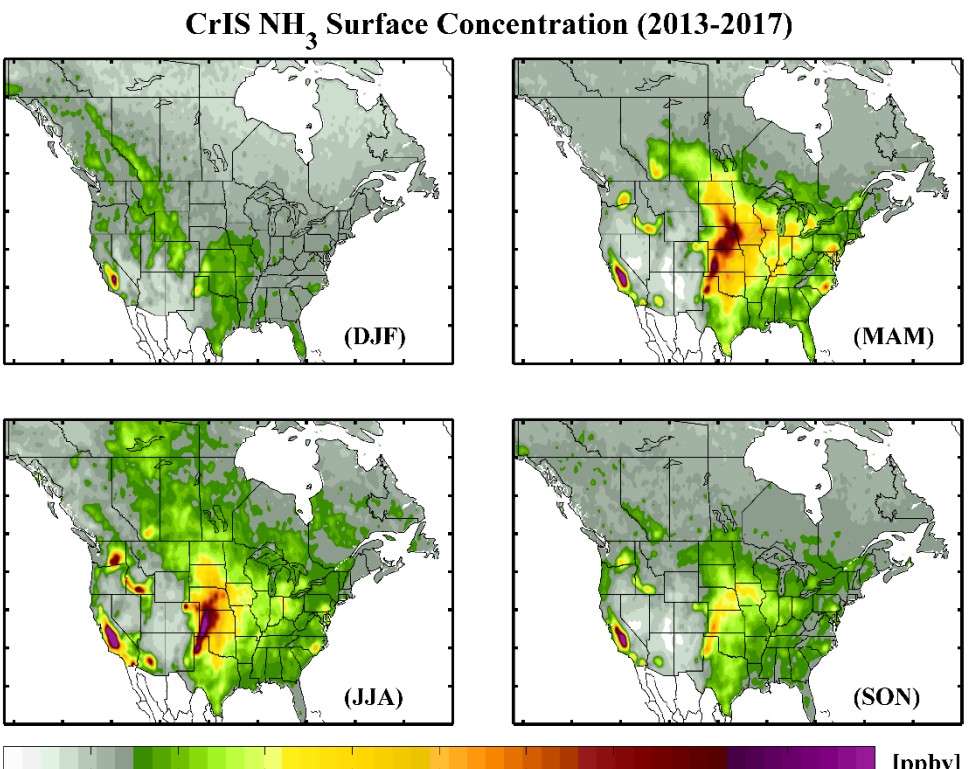

Figure 7. 5-year mean maps of surface NH₃ over North America during the meteorological winter (December-January-February (DJF); spring (March-April-May (MAM)), summer (June-July-August (JJA)), and fall (September-October-November (SON)).



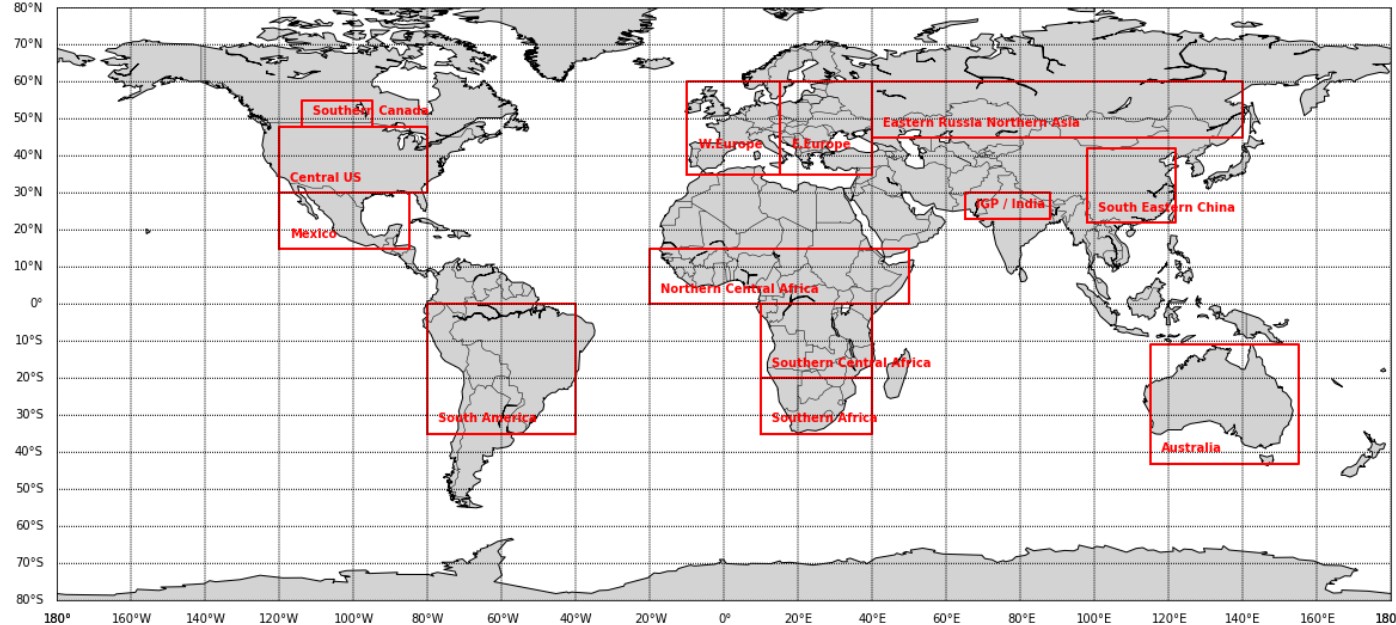

Figure 8. Delimitation of large regions of interest in which CrIS $NH_3$ are grouped together for regional time series analysis.

The time series using 5-years of CrIS data can give additional insight into the change of ammonia over time for various seasons. Time series in CrIS retrieved surface ammonia for the twelve extended regions depicted in Figure 8 are shown in Figure 9. The most salient features in the regional time series reflects the seasonal cycle seen for most regions (e.g. south eastern China, South America), where the ammonia concentrations peak in the warm growing season and are a minimum during the colder season. Some regions also show a double peak in concentration amounts during the growing season (e.g. central U.S.) that can be associated with the large spring time fertilizer or manure application with a second peak in a couple of months later due to increasing temperatures, which can be associated with increased agricultural ammonia volatilization and/or biomass burning. Some regional time series show an increase in peak ammonia concentration amounts with time (e.g. South America), while others show more constant seasonal pattern over the years (e.g. central U.S.). In addition to agricultural practices, there can also be contributions to the atmospheric ammonia amounts due to biomass burning for some of the regions. When wintertime temperatures are near or below freezing there is a decrease in satellite sensitivity to ammonia, which reduces observation density and can create a small high-bias (e.g. southern Canada in 2016 and 2017).



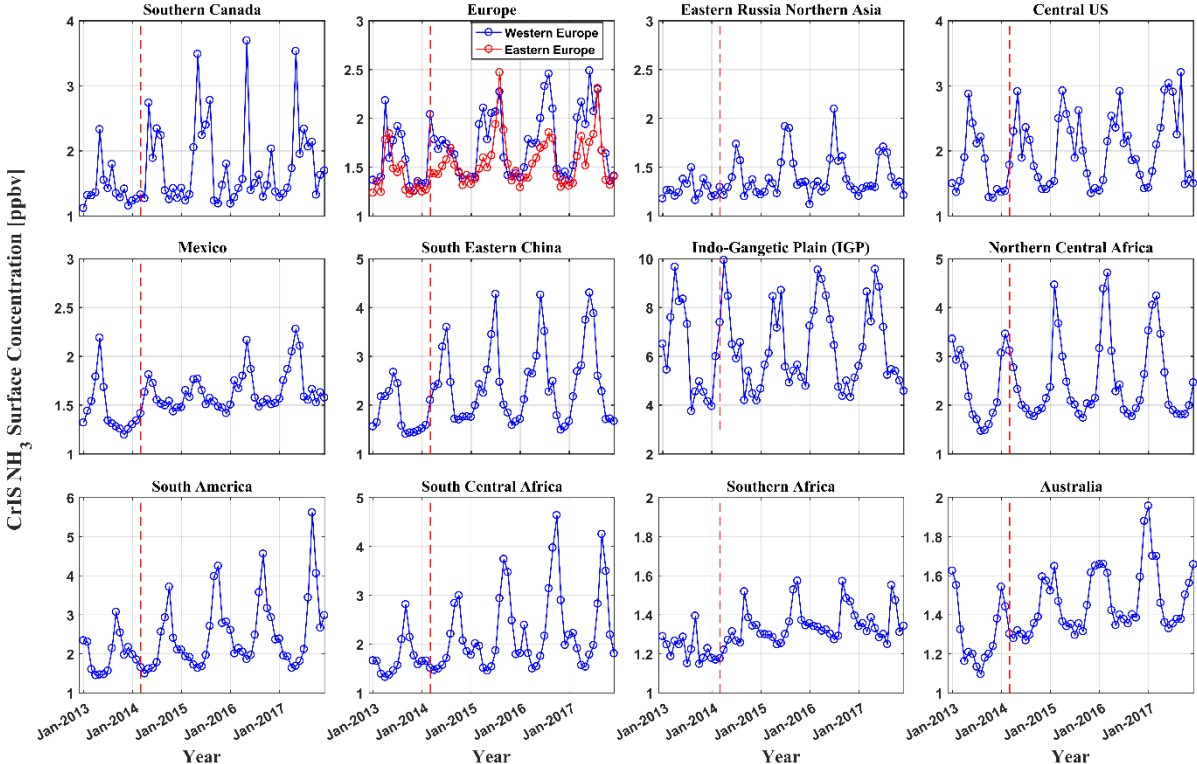

Figure 9. Time series of the CrIS retrieved surface ammonia volume mixing ratios over the regions shown in **Figure 8**. The red vertical dotted line indicates where there was a change in the product used for the input atmospheric state for the CFPR ammonia retrievals.

### 3.1.3   Interannual variability

Even though the lifetime of ammonia in the lower boundary layer is influenced by the concentration of acid gases, especially $SO_x$, one would still expect the mean surface concentrations to be spatially representative of the nearby surface emissions (as shown later in Section 3.2). Figure 10 shows the 5-year (2013-2017) mean plot of CrIS retrieved surface

10 concentrations over North America and the anomalies with respect to this multi-year mean. Many of the regions with elevated concentrations correspond to known agricultural emission zones across North America, such as the Central Valley in California, Washington state, Idaho, the Midwest (e.g. Nebraska), and North Carolina in the U.S.. The agricultural hot spots in Canada are near Lethbridge in Alberta, southern Manitoba, southwestern Ontario, and the St. Lawrence River Valley of Quebec. Going from southern to northern latitudes in North America, the crop-growing season and associated fertilizer

15 application and lower animal densities, especially lower numbers of housed animals with higher emissions than pastured animals, continually decrease, which is reflected in the general decrease in CrIS observed concentrations. The exception is





the contribution of ammonia during wildfires due to both the burning vegetation and volatilized soil nitrogen (e.g. Urbanski, 2004). For example, there was a large number (~385) of forest fires covering ~3.4 million hectares in the Northwest Territories (NWT) during 2014 (2014 NWT Fire Season Review Report), compared to 2013 and 2016 with less fire activity (Munoz-Alpizar et al., 2017). 2017 had relatively larger fire driven emissions in the western and northern parts of the continent, especially in British Columbia, Canada (Chen et al., 2019).

Figure 10. Plots of the CrIS retrieved surface volume mixing ratio concentrations of ammonia. Plot (a) is the 5-year annual average values of the concentrations from 2013-2017. Plots (b) to (f) are annual difference from the 5-year mean for each year from 2013 to 2017.



## 3.2 Model evaluation

Chemical transport models (CTMs) are used for many ammonia related air quality applications such as estimating acid deposition and secondary particulate matter formation, and scenario runs to inform policy development (e.g. Engardt et. al., 2017; Liu et. al., 2019; Makar et al., 2009; Pinder et. al., 2007). Evaluation of the model performance against observations is

a key part of air quality modelling validation that ultimately leads to an improved model. An example of using CrIS $NH_3$ observations for CTM evaluation is provided by Whaley et al., (2018). In that study a newly implemented ammonia bi-direction flux scheme and inclusion of biomass burning into Environment and Climate Change Canada's air quality forecast model, the Global Environmental Multi-scale – Modelling Air quality and CHemistry (GEM-MACH) model (Gong et al., 015; Makar et al., 2015a,b; Pendlebury et al., 2018), were evaluated over northern Canada using CrIS $NH_3$ observations. CrIS $NH_3$

observations have also recently been used to evaluate improvements to the Community Multiscale Air Quality Model (CMAQ) (Pleim et al., 2019). They demonstrated that CMAQ underestimated $NH_3$ concentrations in the spring, but also that CMAQ and CrIS present the same pattern of high $NH_3$ in the California Central Valley, the Snake River Valley and western High Plains, all regions with high soil pH resulting in high $NH_3$ fluxes, suggesting that CMAQ modeling of soil pH and the fluxes dependent on this parameter are reasonably well modeled.

Figure 11 compares CrIS observed $NH_3$ surface concentrations with mean values predicted by GEM-MACH and the corresponding $NH_3$ emissions used by the model. The GEM-MACH-ready hourly gridded $NH_3$ emissions at 10-km resolution over a North American (NA) grid were generated using the SMOKE (Sparse Matrix Operator Kernel Emissions) emissions processing system (Baek and Seppanen, 2018; Zhang et al., 2018). These are based on the 2013 Canada's Air Pollutant Emission Inventory (APEI, accessed 2019), projected 2017 U.S. National Emissions Inventory (NEI) obtained from the U.S.

Environmental Protection Agency's (EPA) 2011 Version 6 Air Emissions Modeling Platforms (AEMP, 2019), and the 2008 Mexican inventory obtained from the EPA's 2007/2008 Version 5 Air Emissions Modeling Platforms (https://www.epa.gov/air-emissions-modeling/20072008-version-5-air-emissions-modeling-platforms). Figure 11a shows the average monthly total $NH_3$ emissions during a two month summer period (July and August) over the GEM-MACH North American grid. The corresponding average $NH_3$ surface concentrations fields predicted by GEM-MACH during July and

August, 2016 is shown in Figure 11b. The spatial distribution of model-predicted anthropogenic $NH_3$ surface concentration aggrees well with the spatial distribution of bottom-up model emissions in Figure 11a. This is expected since $NH_3$ is a short-lived reactive species and high concentrations of $NH_3$ occur mainly in the areas with high $NH_3$ emissions.

In general, the locations of the elevated ammonia "hot-spot" regions in simulated model surface concentration map in Figure 11b match well with those observed by CrIS (Figure 11c). This is seen in the hot-spot regions such as Lethbridge, AB

and southwestern Ontario in Canada, the Central Valley in California, Washington State, Idaho, the Midwest (e.g. Nebraska) and North Carolina in the U.S. The peak values in the hotspots in the upper Midwest and eastern U.S. for these two months in 2016 are generally higher in the model than the satellite observations, whereas in the western part of the U.S. the satellite observations tend to be slightly higher over regions with elevated $NH_3$, most notably over the Central Valley and High Plains.





Similar results (not shown here) are also seen with other chemical transport models (e.g. GEOS-Chem) using the same U.S. EPA emissions inventory. One other potential difference between the model simulations and the satellite observations is the contribution of forest fires to the $NH_3$ concentration amounts, especially at higher latitudes where there are limited agricultural sources. The reason for this potential difference is that the fire emissions were not considered in this GEM-MACH simulation.

5  This demonstrates that these large episodic wildfires in July and August can produce mean bi-monthly concentrations over large regions approaching the elevated anthropogenic agriculture values at lower latitudes, even for the summer of 2016 that was a relatively quiet wildfire season in Canada (Munoz-Aluzar et al., 2017).

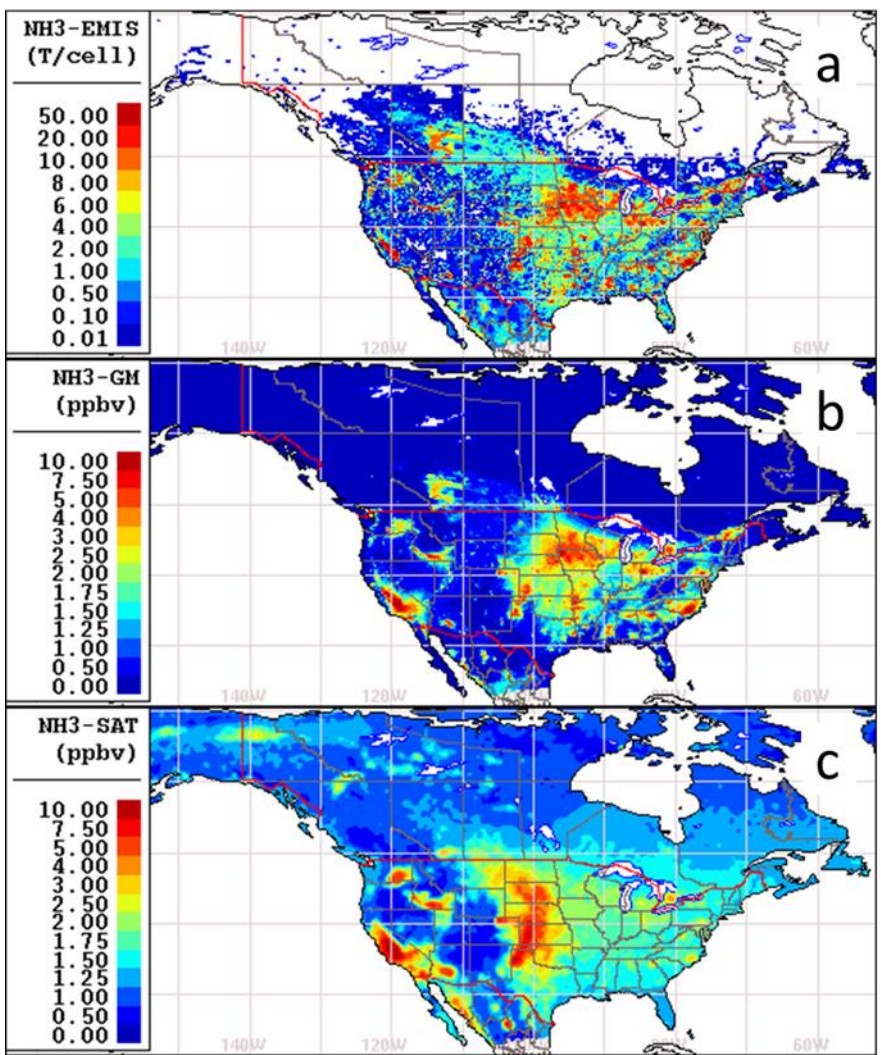

10  Figure 11.  Mean North American agriculture ammonia (a) emissions used by GEM-MACH (b) mean surface $NH_3$ concentrations field for July and August 2016 modelled by GEM-MACH, and (c) corresponding CrIS $NH_3$ surface concentrations field.



### 3.3 Dry deposition of reactive nitrogen

Deposition of basic ammonia and ammonium-containing aerosols on land surfaces leads to acidification of the soil, when

ammonium is oxidized (nitrified) to nitrate ($NH_4^+ + 2O_2 \rightarrow NO_3^- + 2H^+ + H_2O$) (Goulding, 2016). The protons generated from this reaction cause the acidification. The atmospheric deposition of $NH_3$ contributes excessive reactive nitrogen into water that contributes to eutrophication. Kharol et al., (2018) first demonstrated the utility of using CrIS $NH_3$ observations with modelled dry deposition velocities to compute estimates of dry deposition of reactive nitrogen ($N_r$) from ammonia for the 2013 warm (growing, April to September) season over North America. We expanded upon their seasonal analysis to compute

annual estimates of the relative $N_r$ dry deposition flux from $NH_3$ and nitrogen dioxide ($NO_2$) for the years 2013-2017. As described in Kharol et al., (2018), the Ozone Monitoring Instrument (OMI) was used for the $NO_2$ deposition estimates. The spatial patterns of the annual 2013 $NH_3$ dry deposition flux shown in Figure 12 closely resemble the warm season results shown by Kharol et al., (2018) as ammonia has a short lifetime in the atmosphere so the majority of the ammonia deposition occurs on leafy vegetation during the growing season close to agricultural sources. In contrast, the dry deposition of $NO_2$ is

generally associated with emissions from urbanized areas and industrial sources year round, changing only slightly during the warm season. This results in total dry deposition in Canada (excluding Territories) and the U.S. in 2013 from $NH_3$ ($NO_2$) of ~0.75 (0.17) Tg N year$^{-1}$ and ~1.09 (0.91) Tg N year$^{-1}$, respectively, of which ~0.5 (0.1) Tg N warm season$^{-1}$ and ~0.9 (0.4) Tg N warm season$^{-1}$, were deposited in the growing seasons as in Kharol et al., (2018). The 2013 annual ratio maps show $NH_3$ having larger proportion of the ($NH_3 + NO_2$) (~82% and ~55 % over Canada and the U.S.). As shown in Figure 12, 31 out of

the 50 U.S. states are mostly located in central and western U.S. show greater dry deposition rate from $NH_3$ compared to $NO_2$. In contrast, the industrial northeastern states indicate higher dry deposition rate from $NO_2$ than $NH_3$. $NH_3$ is expected to continue to be the dominate source of the reactive nitrogen dry deposition flux over most regions in North America as $NH_3$ emissions are projected to increase in the future (e.g. Bauer et al., 2016; Ellis et al., 2013; Paulot et al., 2013), and because of declining trends in $NO_2$ emissions (e.g. Kharol et al., 2015; Krotkov et al., 2016; Lamsal et al., 2015).



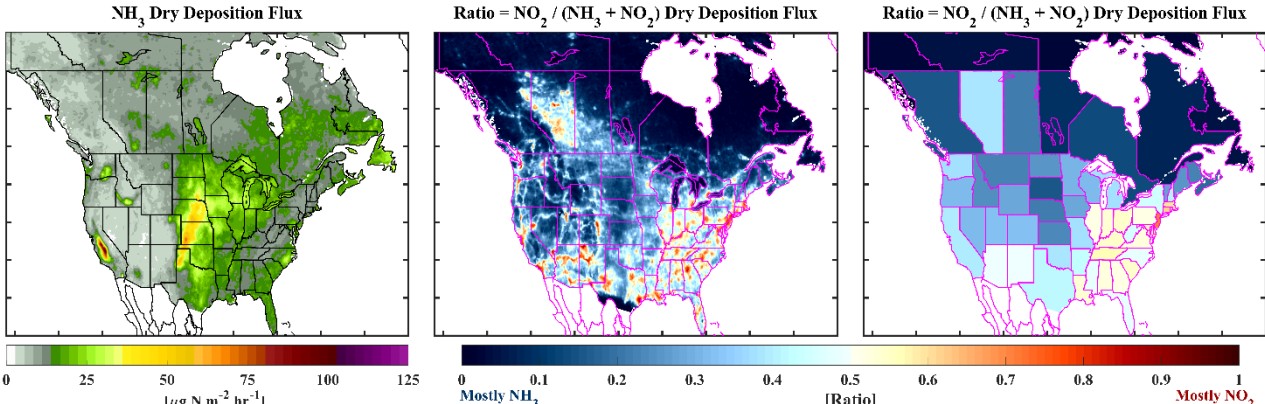

Figure 12. 2013 plots of the CrIS satellite derived dry deposition flux of reactive nitrogen. From left-to-right this figure contains plot of dry deposition from $NH_3$, and the relative ratio of $NO_2$ to ($NH_3 + NO_2$), and this same ratio averaged over geopolitical boundaries.

The five-year annual mean $NH_3$ dry deposition flux for the period of 2013-2017 over North America are shown in Figure 13(a). The year-to-year variability in the $NH_3$ dry deposition flux over North America are shown in Figure 13(b-f). The hotspots in the northern latitudes during 2014 and 2015 are mainly associated with large forest fires that may lead a 2-3 fold local increase relative to background value (Kharol et al., 2018). The $NH_3$ dry deposition flux hotspots evident in the agricultural states of central U.S., and the Canadian provinces of Alberta, British Columbia and Saskatchewan during 2017 (Figure 13f) are mainly due to the combined effect of forest fires and a warmer than average summers as shown in the CrIS $NH_3$ concentrations maps in Figure 10. The annual average and variability in ammonia dry deposition of reactive nitrogen over Canada and the U.S. is ~0.8 ± 0.08 Tg N year$^{-1}$ and ~1.23 ± 0.09 Tg N year$^{-1}$, respectively. Note that there will be a significant contribution and variability from large forest fires in northern latitudes as seen in Figure 13.



**NH₃ Dry Deposition Flux**

**NH₃ Dry Deposition Flux Difference**

**NH₃ Dry Deposition Flux Difference**

**NH₃ Dry Deposition Flux Difference**

**NH₃ Dry Deposition Flux Difference**

**NH₃ Dry Deposition Flux Difference**

Figure 13. Plot (a) is the 5-year average (2013-2017) of the ammonia dry deposition flux over North America. Plots (b) – (f) are the differences in the NH₃ dry deposition flux for each year from the 5-year mean shown in (a).

## 3.4 Emissions estimate for a concentrated agricultural region

Another application of CrIS ammonia observations are emission estimates. Emission inventories are traditionally built from the bottom up, using emission factors and source locations to construct a complete inventory (e.g. see Appendix D). This process is very labor intensive, which means that the inventories are often released somewhat infrequently, with gaps of a few years between releases. Furthermore, the inventories can be incomplete or inaccurate due to a lack of knowledge on source





locations, magnitudes, and temporal variations. This is particularly true for farm based emissions that require complex data about farm activities, which can only be obtained with complex farm surveys typically conducted sporadically. Top-down satellite observations can be used to provide another source of emissions information and to supplement the inventories with more detailed information on hotspot locations and temporal variations (e.g. seasonal and inter-annual variations). The use of

CrIS $NH_3$ concentration observations with corresponding wind information to derive emissions have been demonstrated for wildfire sources (e.g. Fort McMurray forest fires that occurred in 2016 in Alberta, Canada (Adams et al., 2019)), and globally from localized industrial sources (e.g. fertilizer factories) (Dammers et al., 2019). Here we demonstrate the potential to use CrIS $NH_3$ observations over extended agricultural area by adapting a similar emission estimate technique as used by Dammers et al. (2019) to estimate $NH_3$ emissions over an agricultural region in Lethbridge, Canada with many Concentration Animal

Feeding Operations (CAFOs). A detailed explanation on the procedure for the satellite derived ammonia lifetime and emission estimates is provided in Dammers et al. (2019). The initial CrIS dataset is first filtered for observations with a quality factor flag of 5 and DOF filter $\geq$ 0.8. Next, we removed the influence of nearby forest fires following Dammers et al., (2019). The remaining CrIS satellite observations were matched in space and time with the ERA-Interim wind fields, and using that information the individual measurement locations are rotated according to their wind direction about a reference point in order

to align their values (Pommier et al., 2013).

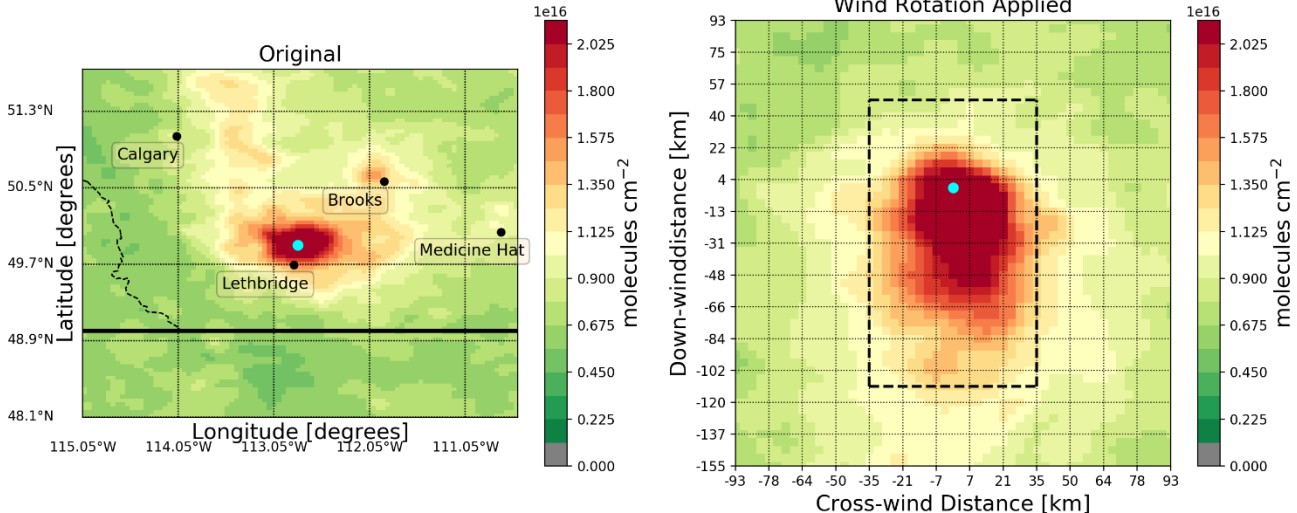

Figure 14. Plot of the CrIS-$NH_3$ total column observations over Lethbridge, Alberta, Canada. The left plot shows the distribution of $NH_3$ for the surrounding region. The black dots indicate nearby population centres for easy reference. The right hand plot shows the same set of observations for a smaller region centred on Lethbridge, but now redistributed by wind

direction showing the upwind and downwind total columns. The blue circle shows the location of the reference point used in the wind rotation, and corresponds to the geographic center of emissions.

Figure 14 shows the 5-year (2013-2017) unrotated mean total column concentrations in southern Alberta (left) and the rotated total column (right) concentrations around Lethbridge, Alberta for the same period. The concentrations peak near the Lethbridge location, with the peak transported downwind due to advection. The lifetime, plume-spread, background and





emission enhancement parameters are then estimated using a 2D exponentially-modified Gaussian (EMG) plume model. Over this region we only use warm season observations to optimize for favourable measurement conditions (6-months from April-September) and obtained a of 37.35±6.3 kt warm season$^{-1}$, with a plume spread of 18.9±0.2 km, and a lifetime of 2.66±0.04 hours. Assuming that emissions are constant in time the annual emissions would be 74.7±12.6 kt yr$^{-1}$. However, emissions in

this region are expected to vary, especially on a monthly/seasonal basis. The 2013 Canadian Ammonia Emissions from Agriculture Indicator (AEAI) monthly emission inventory (Sheppard and Bittman, 2016) states 30% of all emissions are assume to take place in the cold season (October-March) and 70% in the warm season (April-September). Therefore, the annual emission total has to be adjusted by a factor of 1/0.7 = 1.43, which makes the adjusted emission total of 53.4±9.0 kt yr$^{-1}$. Similarly, the diurnal emission profile can be approximated by using the diurnal emission profile for livestock that is used to

prepare emissions for the GEM-MACH model. Most of the emissions around Lethbridge are from livestock, which has a peak in the morning to middle of the day following cattle activity including feeding and excretion and the increasing surface temperatures (Denmead et al., 2014; Van Haarlem et al., 2008), and are thus emitted before the satellite overpass time (~1:30 local time). Thus, the inventory value from an hour before the overpass (12:30 local time) is used to adjust the emissions to daily averages by a factor of 1/1.44 leading to a final annual emission total of 37.1 ± 6.3 kt yr$^{-1}$. Figure 15 show the CrIS

estimated emissions compared to the AEAI, Canada's Air Pollutant Emission Inventory (APEI) (APEI, 2017) (see Appendix D), and Global Hemispheric Transport of Air Pollution (HTAPv2) gridded emissions inventory (https://edgar.jrc.ec.europa.eu/htap_v2/). The Lethbridge regional emissions are summed over a box with a bottom-left corner of (49.35ºN, 113.31ºW) and a top-right corner of 50.28ºN, 111.10ºW, which corresponds to an elevated emission region in the inventories.

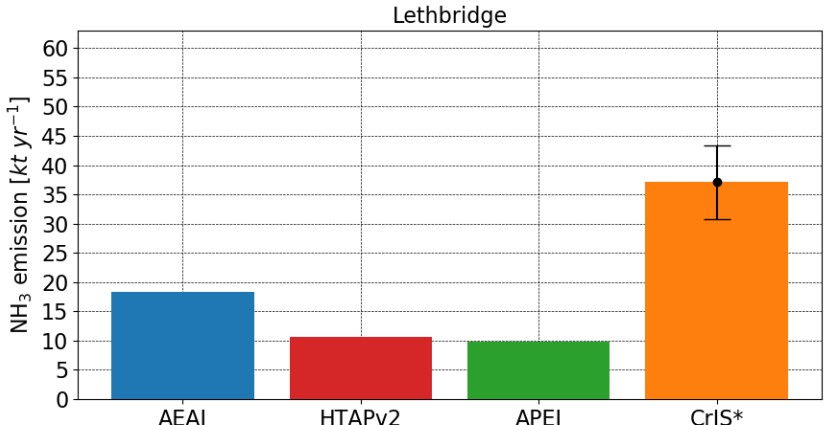

Figure 15. The bars show the Lethbridge regional emissions as in the AEAI (blue), HTAPv2 (red), and APEI (green) inventories summed over a box with a bottom-left corner of (49.35º, -113.31º) and a top-right corner of (50.28º, -111.10º). The orange bar shows the CrIS estimated emissions using the free fitting algorithm and adjusted for diurnal and seasonal variability.

The vertical error bar shows the uncertainty of the CrIS estimate that includes the uncertainty in the fit, the total columns, and uncertainties due to the meteorology.





In addition to the annual total, emissions are estimated using 5-year monthly observations and a moving window of 31 days in 1 day increments. Figure 16 shows the results of the emission estimates, compared to the AEAI 2013 monthly emission inventory (blue). The orange lines and red dots show the results when applying the plume fitting algorithm to the CrIS monthly dataset and daily sets (31-day moving window), respectively. While some months might have enough

5   information available in the mean total column fields, this is not true for each of the months as indicated by the relatively noisy results outside of the warm season. To improve the stability of the fitted results a lifetime of 2.65 hours and a plume spread of 19 km as obtained from the 5-year analysis above, were used here to estimate the CrIS derived monthly and daily emissions in Figure 16. This lifetime and plume spread distance were derived based on 5 years of CrIS data. This type of analysis can be used to help better capture the warm season timing of emissions over this region.  The general overall seasonal changes in the

10   spring and summer are similar between the CrIS derived emissions and the AEAI inventory with peak emissions over Lethbridge in May springtime, but CrIS is showing nearly double the amount compared with AEAI.  In this example, CrIS is not showing the smaller secondary fall peak (September-October) that is seen in the AEAI inventory.

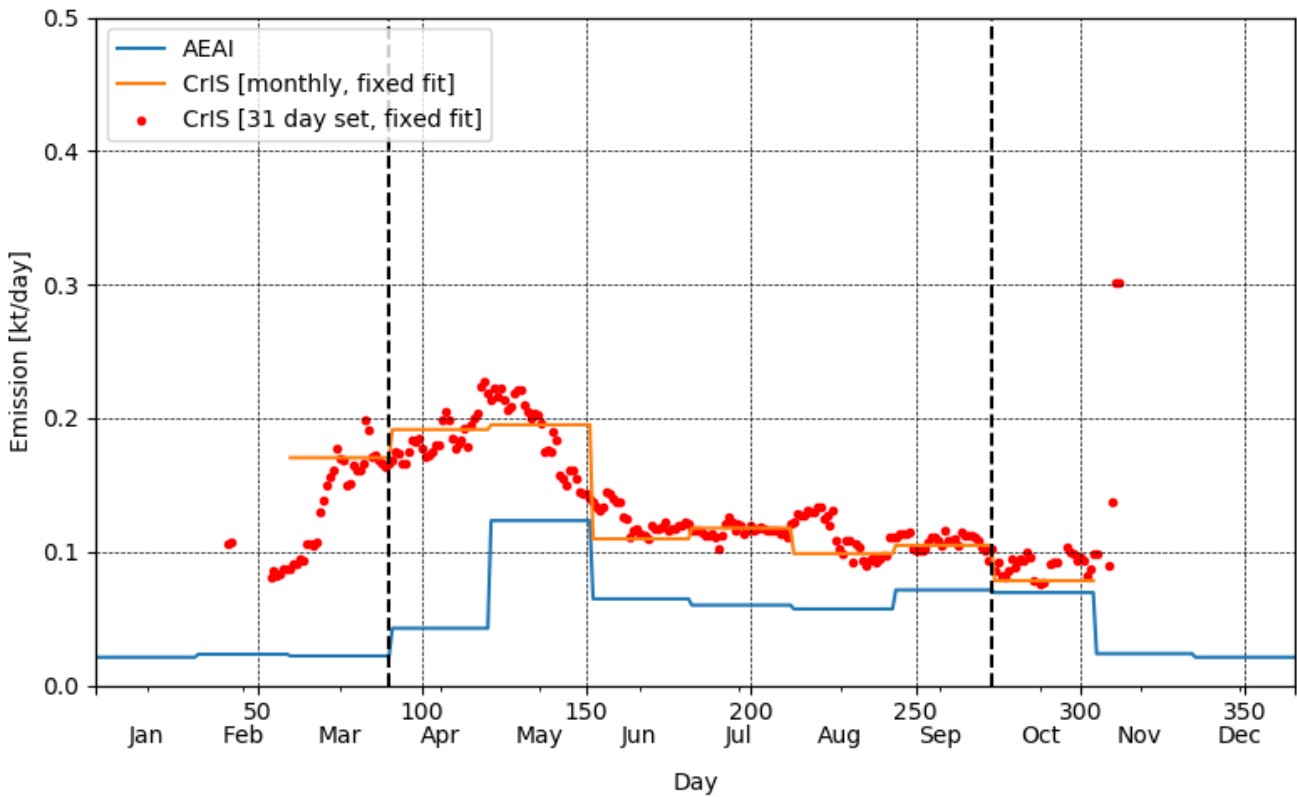

Figure 16.  Plot showing the monthly emissions. The monthly emissions following the Canadian AEAI 2013 inventory are shown in blue, which are summed over the Lethbridge region (a box with a bottom-left corner of (49.35°, -113.31°) and a top-





right corner of (50.28º, -111.10º)). The orange line shows the CrIS monthly estimated emissions using a lifetime of 2.65 hours and a plume spread of 19 km, both of which are derived from the 5-year data record. The red dots are the corresponding CrIS derived emissions in daily increments that are derived using a 31-day interval (with 5 years, namely 2013-2017). The vertical dashed black lines indicate the beginning and end of the warm season over this region.

## 4   Conclusions

Satellite observations of lower tropospheric ammonia are a relatively new development with the initial proof of concept in the past decade (Beer et al., 2008), thus, there is a great potential for advancements in the retrievals and exploration of new applications. Presented here is an overview of CrIS' ammonia data product highlighting its current capabilities to observe lower tropospheric ammonia with sample applications for monitoring, model evaluation, dry deposition, and emission estimates. The CrIS daily observations demonstrate the influence of meteorology on the spatiotemporal variability of ammonia. These examples show the transport of ammonia concentrations from nearby agriculture sources as well as from fire emissions. Averaging these daily observations over longer time-periods (e.g. monthly, seasonal, and annual) and gridding and oversampling (to yield Level 3 products) illustrates the spatiotemporal variability of ammonia at various timescales. These results demonstrate CrIS' ability to observe regional changes in ammonia concentrations due to agricultural practices, such as spring maximum values over agricultural regions when ammonia is released into the air from the fertilizing of crops. Also shown is the importance of episodic wildfire emissions in the more wildfire active months, especially in regions where there is little or no agriculture sources such as the northern latitudes in North America during July and August.

Initial comparisons of CrIS $NH_3$ satellite observations with GEM-MACH air quality model simulation in summer 2016 show that in some regions there is general agreement on the spatial distribution of the anthropogenic hotspots, while other areas are markedly different and will need further investigation. For this summer period, the model tends to have higher peak values in the eastern U.S., whereas the satellite tends to have larger peak values in the western half of the U.S. As the CTM runs only have anthropogenic emission sources included, we can see that the impact of large summertime wildfires at higher latitudes on the 2-monthly mean concentration levels over large regions can be significant, and can approach the values of agriculture hotspots at lower latitudes.

Expanding on the initial 2013 growing season results from Kharol et al, (2018) we show annual dry deposition rates of nitrogen from ammonia for 5-years from 2013 to 2017 over North America. CrIS satellite derived values show the annual average and variability in the dry deposition of reactive nitrogen from ammonia over Canada and the U.S. of ~0.8 ± 0.08 Tg N year[-1] and ~1.23 ± 0.09 Tg N year[-1], respectively. When combining with OMI-derived $NO_2$, the 2013 annual ratio shows $NH_3$ accounting for ~82% and ~55 % of the combined reactive nitrogen dry deposition from these two species over Canada and the U.S. CrIS satellite observations are also used to derived agricultural emissions over the CAFO dominated region of Lethbridge, Alberta, Canada for a five-year period (2013-2017). The satellite-derived annual emissions in the Lethbridge region are 37.1 ± 6.3 kt/yr, which is at least twice the value reported emissions in current Canadian and Global bottom-up



emission inventories over this region. Furthermore, we demonstrated the potential of using a time series of the satellite-derived emissions to evaluate the seasonal temporal emissions profile used in bottom-up inventories over an agriculture hotspot.

Additional application such as using the CrIS CFPR products for model inversions and data assimilation (Lonsdale et al., 2019; Li et al., 2019) are currently being explored, which take advantage of the averaging kernels and error covariance matrix provided in the CrIS retrieved product (e.g. observation operator), to provide top-down constraints on the ammonia emissions. Additionally, we will continue to refine and validate the CFPR algorithm and product. Some of these potential efforts include: (i) accounting for cloud-free pixels that have no information (no ammonia signal in the spectra) in the CrIS composite (Level 3) products globally for the entire dataset; (ii) investigating retrievals over ocean and elevated concentration values over some deserts and high elevations wintertime conditions (e.g. North American Rockies); (iii) investigating the potential enhancements to the *a priori* profiles and constraints used in the retrievals; (iv) and validating CrIS NH$_3$ night time observations against available ground-based observations.

## Acknowledgements

We would like to acknowledge the NOAA Comprehensive Large Array-Data Stewardship System (CLASS) (Liu et al., 2014), with special thanks to Axel Graumann (NOAA), for providing the CrIS Level 1 and Level 2 CrIS REDRO and NUCAPS input atmospheric state data. We thank Denis Tremblay (Science Data Processing, Inc.) for providing value insights on the performance and characteristics of the CrIS instrument, and Cristen Adams (EMSD, Government of Alberta) for detailed discussions on the fire observations. We are grateful to Leiming Zhang (ECCC) for his helpful discussions on dry deposition. Karen Cady-Pereira (AER) contribution was supported through funding from NASA grants NNH15CM65C, 80NSSC18K1652, and 80NSSC18K0689. Matthew Alvarado and Chantelle Lonsdale (AER) were supported by NOAA Climate Program Office grants NA13OAR4310060 and NA14OAR4310129 and NASA Applied Science grant 80NSSC19K0190. We would like to thank Nick Krotkov (NASA) for his support with the OMI NO2 product.

*Author contributions.* MS and KCP developed the CrIS CFPR Level 2 ammonia product. MS, ED, JT, YGM, AK, and SK processed the Level 2 product and developed the CrIS Level 3 gridded ammonia product. ED performed the emission calculations, and SK provide the dry deposition results. JZ, MM, VSJ, and QZ produced the GEM-MACH model results. SB provided the Canadian AEAI emissions and contributed to the analysis of the satellite-derived emission estimates. MS, ED, KCP, SK, CM, CS, MA, CL, and DG contributed to the analysis of the CrIS satellite retrievals and applications.



*Competing interests*. The authors declare that they have no conflict of interest.

*Code and data availability*. The CrIS NH$_3$ Version 1.5 is produced at Environment and Climate Change Canada (ECCC) and Atmospheric and Environmental Research (AER) using the CFPR algorithm, which is the research version of the near future

5 NASA operational CrIS NH$_3$ retrieval. The CrIS CPFR Version 1.5 ammonia data is currently available from Environment and Climate Change Canada (ECCC) upon request (Mark.Shephard@canada.ca). Python/Matlab code used to create any of the figures is available on request. We use the OMI operational NO$_2$ standard product (SP), version-3 (https://disc.gsfc.nasa.gov/datasets/OMNO2_V003/summary).



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

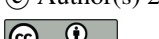

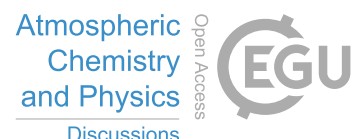

Appendix/Supplemental

## Appendix A: Quality Flags of Version 1.5

5    Here are the quality flags specified for Version 1.5. The quality flags become more conservative with increasing values, thus, they are applied as greater than or equal to the level you want to use.

**Table 1 Quality Flags for Version 1.5**

| Quality Flag | Description | Details |
|---|---|---|
| -1 | Retrieval did not converge | Flag indicating that the retrieval did not converge. Often these are not written into the product files. |
| $\geq 1$ | Retrieval converged | Flag indicating that the retrieval converged is specified number of iterations. |
| $\geq 2$ | Large outlier flag | Quality flag 1 & profile retrieved value less than 200 ppbv. |
| $\geq 3$ | Retrieval quality flag | Quality flag 2 & chi-squared less than 20 |
| $\geq 4$ | More conservative retrieval quality flag | Quality flag 3 & signal-to-noise $\geq 1$ and thermal contrast $> 0$ |
| $\geq 5$ | More conservative retrieval quality flag with information content | Quality flag 4 & also filtered to have a minimum degree-of-freedom for signal of 0.1 |

## Appendix B: Calipso





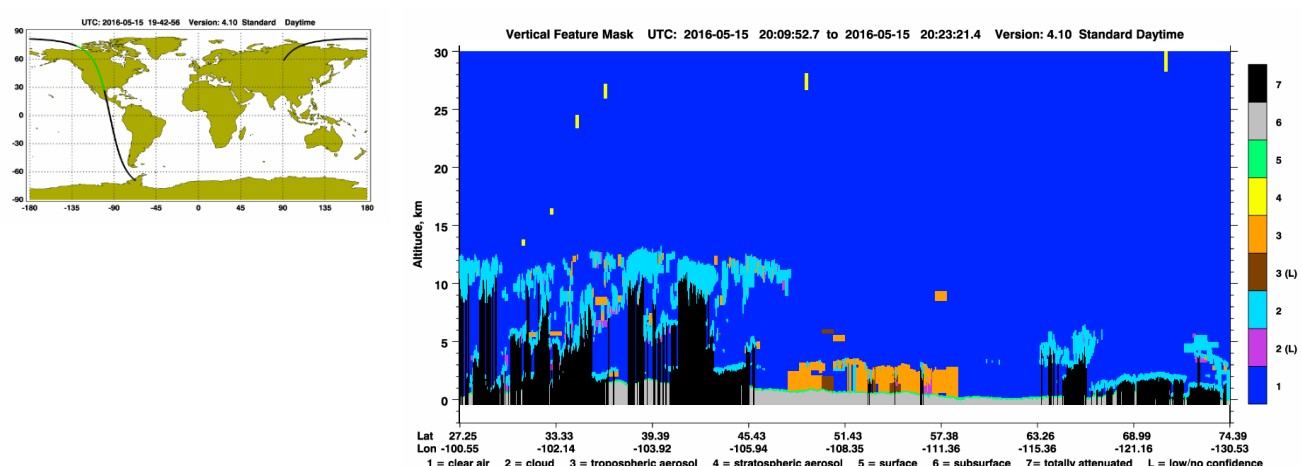

**Figure A 1. (CALIPSO plots obtained from https://www-calipso.larc.nasa.gov/data/BROWSE/production/V4-10/2016-05-15/)**

### Appendix C: Total Column Figures

5         As a reference, this section contains total column maps that correspond to annual and seasonal surface ammonia over North America shown in the main part of the manuscript.



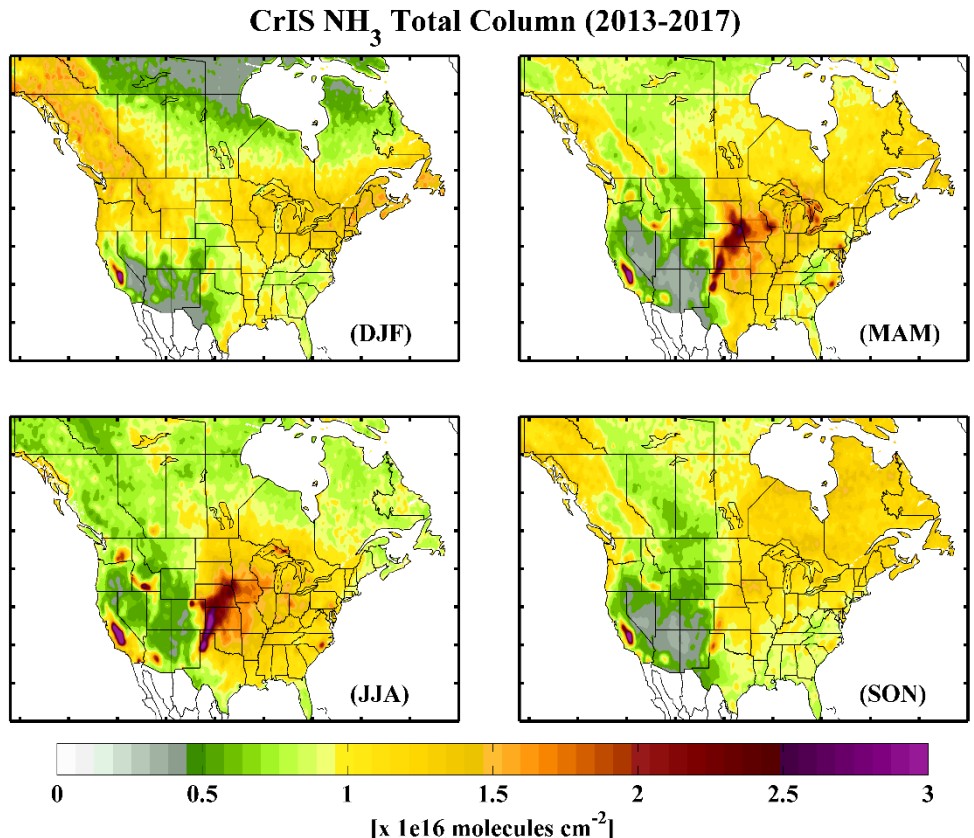

**Figure A 2.** Similar to Figure 7 but for total column amounts.

**Figure A 3.** Similar to Figure 10 but for total column ammonia amounts

.

**Appendix D:  Brief description of Canada's Air Pollutant Emission Inventory (APEI)**

The AEAI inventory calculation (from agricultural sources) is a multi-stage process involving three sets of information. The inventory is built on very detailed census data on animals for each census district collected annually by Statistics Canada. The data on fertilizer use, including forms of nitrogen, is provided by the fertilizer industry on a provincial basis. Ammonia emissions are strongly influenced by farming practices such as manure handing systems or fertilizer application method. The practices data were acquired by farm surveys, targeted to ammonia with an emphasis on timing of





practices, across the main livestock sectors across the 12 key ecoregions across Canada. There was a multisector survey in 2005 that, targeting ammonia related practices from feed quality, to housing and storage facilities to land application practices and grazing management. The important beef sector survey was updated in 2011 to capture large changes in practices. A new pig survey will be conducted this year. A separate survey for fertilizer practices relating to ammonia was

5    conducted in 2006

by the polling company IPSOS.

Emission factors for the particular farm practices were obtained from scientific studies conducted in Canada and elsewhere. Some published models for emissions were used, and where possible tested with Canadian data. The emission

10   factors were adjusted for ambient temperatures relating to the practices, the regions and the time of the practices. The manure application emissions were also adjusted for the probability of rainfall. The emission data was granulated to a 50x 50 km grid by ECCC and a finer grid is being contemplated. Note that in some cases, notably where there are few operations, the data is

averaged over a larger areas with more operations to ensure confidentiality for the farms.