# Peer review of "Ammonia measurements from space with the Cross-track Infrared Sounder (CrIS): characteristics and applications"

_Atmospheric Chemistry and Physics, 2019_

## Referee Comment (RC1) · Anonymous Referee #1 · 11 Oct 2019

The manuscript by Shephard et al. entitled "Ammonia measurements from space with the Cross-track Infrared Sounder (CrIS): characteristics and applications" presents the characteristics of the CrIS NH3 Fast Physical Retrieval (CFPR). Several examples are shown and applications are detailed (monitoring, model evaluation, dry deposition estimates and emission estimates). For several applications, the results have already been published and updated numbers are provided taking advantage of the newly available 5-year dataset.

The paper is well written and the figures provide nice illustrations of the applications. However, as the retrieval approach as already been published, as well as a substantial part of the results, I wonder if the title should not be adapted to reflect this (e.g., by adding "update of" before characteristics and applications?).

I recommend the paper to be published after the consideration of the following important points:

1. Some numbers/statistics to better characterize the CrIS dataset are missing. Firstly on the DOFS: How are they distributed? Could you provide a representative distribution? What are the proportion of the CrIS measurements characterized by a DOFS>0.5 for example? Secondly, the paper misses a discussion on the bias introduced by the pre- and post-filtering of the data. First of all, the fact that the entire dataset is filtered using an "estimated ammonia spectral signal" will bias high any type of average. This should be specified and commented in Section 2; I suggest you to move and expand the last paragraph of Section 3.1.1 in Section 2. As an illustration of this, it can be observed in Figure 7 that the surface concentrations are very high in terms of ppb- (5-year average >1-2 ppb in remote areas). Figure 9 also shows relatively high winter values, likely due to the data filtering (and the fact that low $NH_3$ values are filtered out).
Interesting numbers to add in the study would also be (1) the proportion of CrIS observations processed and (2) the number of observation per quality flags to assess their impact on the data availability. In addition, the number of observations used to make the map of each subpanel of Figure 7 (and Figure A2) would be useful to exclude any sampling bias.

2. As stated by the authors, the DOFS are generally between 0.1 and 1 (rarely above 1 if I understood it well). Is it therefore relevant to present surface concentrations? Why don't you present integrated columns when you show CrIS $NH_3$ distributions?
The total errors shown in figures 3 and 4 also strongly encourage the use of the total columns.
Regarding this issue, could you comment the differences that can be observed between the 5-year averaged surface concentration distribution shown in Figure 5 and the 5-year averaged total columns distribution shown in Figure 1 of Dammers et al., ACP 2019 paper (apparently made with the same dataset).
In the model comparison, why not comparing directly the total columns instead of surface concentrations?
I am also curious to have a look at a global distribution made only with observations characterized by a DOFS above 0.5. Would you mind providing such a figure?
Finally, when calculating the estimated emissions based on CrIS NH3 measurements, you only keep the observations with a DOFS above 0.8. How this threshold has been defined? Why not always using this threshold? How does it affect your results?

3. I was surprised to note that the authors do not refer to relevant work made on the remote sensing of atmospheric $NH_3$ using the IASI satellite, such as the first global distribution provided from space measurements presented in Clarisse et al., NatGeo 2009 (doi: [10.1038/ngeo551](10.1038/ngeo551)) and the first estimation of $NH_3$ emissions from industrial and agricultural sources in Van Damme et al., Nature 2018 (doi: [10.1038/s41586-018-0747-1](10.1038/s41586-018-0747-1)). At the very least, these two papers should be cited.

Minor comments:

- Several papers are not referenced in the bibliography:
  Divarkarla et al., 2014 ; Li et al., 2019 ; Ellis et al., 2013 ; Losdale et al., 2019
- Units are generally missing in the captions of the figure, I would add them
- Consistency in the writing of "hotspot" (currently, we have in the paper: hotspots (e.g., p1), "hotspots" (e.g., p8), hot spots (e.g., p9), hot-spot (e.g., p17) and "hot-spot" (e.g., p17))
- P2, L26: change Clarisse et al., 2010 to Clarisse et al., 2009
- P3, L15-18: What is impact of the change in Level 2? From what can be seen in Figure 9, it looks not negligible for some regions. Could you comment on this in the text?
- P3, L26-30: Have you changed the pre-filtering following that update? It should be specified in the text.
- P14, Figure 8: this figure could be moved to the supplementary materials (or at least reduced in size)
- P15-16, Figure 10: What is striking in this figure is the increase in 2017. While this is mention later in the text, the description of that figure does not mention it.
- P17-18, Figure 11: Lethbridge is highlighted as a place where the model match well the satellite data. However, if you look at the southern part of Alberta, this is not really the case. Could you comment on this? While I agree that overall, there is a good match between the model and the satellite distributions, I miss some discussion on the large regions where the model disagree with CrIS. Eastern US is an important example to discuss but also Mexico (while introduced when listing the emission inventories, it is not discussed at all in the text).
- P23, Figure 15: Having in mind the data filtering ("cloud screening", DOFS>0.8, only warm season considered) and the different adjustments applied on the calculate emissions, I found the error bars presented in the figure not really meaningful and think that it could be misleading (even if the caption lists what is included in the uncertainty estimate).

Technical corrections:

- P2, L6: $NH4^+$ -> $NH_4^+$
- P2, L18: add a comma after "(hours to a day)"
- P3, L16: Cross-Track -> Cross-track
- P3, L18: CPFR -> CFPR
- P3, L31: Since he -> Since the
- P4, L21: 1to -> 1 to
- P4, L22: elevationsin -> elevations in
- P5, L15: a bracket is not closed -> add ")" between "error" and "are"
- P9, L18: consider rephrasing "(do that there is bi-directional flow)"
- P12, L7: norther states -> northern states ?

- P17, L23: two month summer period -> two-month summer period
- P17, L16-17: consider rephrasing: "The GEM-MACH-ready hourly gridded NH3 emissions at 10-km resolution over a North American(NA) grid […]"
- P18, L7: "Munoz-Aluzar" does not correspond to what is in the reference list ("Munoz Alpizar")
- P19, L6-7: Consider rephrasing "The atmospheric deposition of NH3 contributes excessive reactive nitrogen into water that contributes to eutrophication. "
- P19, L20: "50 U.S. states are mostly located in central and western U.S. show greater […]" -> "50 U.S. states (mostly located in central and western U.S.) show greater […]"
- P22, Figure 14: the caption misses the time period (and units) of the data used
- P23, Figure 15: CrIS* -> CrIS
- P24, Figure 16: rephrase "Plot showing the monthly emissions." E.g.: "Time-series of the monthly $NH_3$ emissions (kt/day) over Lethbridge (AB, Canada)."
- P37, Figure A 3: I would saturate a bit more the difference plot -> colorbar from -1 to 1 e16 for example?
- P32, L1: PM2:5 -> PM2.5
- P32, L15: retrievalfrom -> retrieval from
- P33, L4: reference "2014 NWT Fire Season Review Repot" to be revised
- P38: consider rephrasing sentence L1-3

---

## Referee Comment (RC2) · Anonymous Referee #3 · 15 Oct 2019

The manuscript entitled "Ammonia measurements from space with the Cross-track Infrared Sounder (CrIS): characteristics and applications" describes the strategy of ammonia retrieval from CrIS on Suomi-NPP and availabilities of the product. This paper investigates the capabilities of the product for monitoring, model evaluation, dry deposition, and emission estimates. This work is quite important and largely contributes to the atmospheric chemistry community. I recommend publishing the paper after addressing several questions and comments.

Major comments 1) Cloud filtering is mentioned in section 3.1.1. Why didn't you use the VIIRS data in the current system? In Fig.2, several scans over clouds are not

eliminated. The cloud filtering described in the paper seems not so accurate and likely occur some bias to ammonia concentrations. The author should add some evaluations of cloud filtering. In addition, I would suggest that the description of the cloud filtering strategy is moved to section 2. 2) Page 18, line 2: Why did you select that two months, July and August? How is the consistency in the other seasons that there are no obvious wildfires? The figure for the difference between Fig. 11a and 11b is also helpful.

Minor comments 1) Figure 1: The account for the circle and bar plots in the right figure should be added in the caption. 2) Page 4, line 22: "elevationsin" should be "elevations in". 3) Page 5, line 15: ")" corresponding to "(includes…" is missing (after "(or smoothing) error"?). 4) Figure 5: Some other papers (e.g. Van Damme et al., 2014) and Warner et al. (2016)) reported the high concentration in Siberia and Alaska. I would recommend expanding the latitudinal area to around 70N. If there aren't significant high values, it is also valuable information. 5) Figure 5: Are the plots only over land? If so, it should be added to the body text or caption. 6) Page 9, line 30: "SCCP" should be "ISCCP". 7) Page 12, line 7: "norther" should be "northern". 8) Page 17, line 8: "Gong et al., 015" should be "Gong et al., 2015". 9) Page 17, line 26: "aggrees" should be "agrees".

---

## Referee Comment (RC3) · Anonymous Referee #4 · 29 Oct 2019

This manuscript by Shephard et al. titled "Ammonia measurements from space with the Cross-track Infrared Sounder (CrIS): characteristics and applications" is well written and very easy to follow. They describe well about the CrIS NH3 product and demonstrate the capabilities of this product for multiple applications, such as model evaluation and emissions estimates. The manuscript does a thorough job of describing the CrIS NH3 CFPR algorithm and its various components. The demonstration of the applications of this dataset is also comprehensive and convincible. Overall, I think it is appropriate for publication after minor revision. I have a few specific comments below.

Page 3, Line 24-25: "higher concentration . . . near the surface. This is demonstrated

[Figure]

later in Section 3.2 with model emissions and corresponding simulated surface concentrations." I think this sentence is not appropriate. The higher concentrations in the model cannot demonstrate the findings from the CrIS since the emissions always emit at the surface level in the model and thus will of course get higher simulated surface concentrations.

Page 24: The paragraph of introducing the "31-day moving window" is not very clear to me. What is the purpose of developing these daily sets? Why do they choose 31 days? Also, I'm not sure how it helped "better capture the warm season timing of emissions". It may be better for readers to understand if the authors could answer all the questions above in the manuscript.

Typo: Page 3, Line 31: Since he -> Since the. Page 5, Line 6: Figure A 1 -> Figure A.1.

---

## Author Comment (AC1) · 3 Jan 2020

Responses to the reviews for the manuscript titled, "Ammonia measurements from space with the Cross-track Infrared Sounder (CrIS): characteristics and applications".

We would first like to thank the Reviewers for taking the time to review the manuscript as their comments and edits strengthened the paper.  Below you will find responses to each of their comments.

**Reviewer #1**

The manuscript by Shephard et al. entitled "Ammonia measurements from space with the Cross-track Infrared Sounder (CrIS): characteristics and applications" presents the characteristics of the CrIS NH3 Fast Physical Retrieval (CFPR). Several examples are shown and applications are detailed (monitoring, model evaluation, dry deposition estimates and emission estimates). For several applications, the results have already been published and updated numbers are provided taking advantage of the newly available 5-year dataset. The paper is well written and the figures provide nice illustrations of the applications. However, as the retrieval approach as already been published, as well as a substantial part of the results, I wonder if the title should not be adapted to reflect this (e.g., by adding "update of" before characteristics and applications?).

I recommend the paper to be published after the consideration of the following important points:

1. Some numbers/statistics to better characterize the CrIS dataset are missing. Firstly on the DOFS: How are they distributed? Could you provide a representative distribution? What are the proportion of the CrIS measurements characterized by a DOFS>0.5 for example? Secondly, the paper misses a discussion on the bias introduced by the pre- and post-filtering of the data. First of all, the fact that the entire dataset is filtered using an "estimated ammonia spectral signal" will bias high any type of average. This should be specified and commented in Section 2; I suggest you to move and expand the last paragraph of Section 3.1.1 in Section 2. As an illustration of this, it can be observed in Figure 7 that the surface concentrations are very high in terms of ppb- (5-year average >1-2 ppb in remote areas). Figure 9 also shows relatively high winter values, likely due to the data filtering (and the fact that low $NH_3$ values are filtered out).Interesting numbers to add in the study would also be (1) the proportion of CrIS observations processed and (2) the number of observation per quality flags to assess their impact on the data availability. In addition, the number of observations used to make the map of each subpanel of Figure 7 (and Figure A2) would be useful to exclude any sampling bias.

*The reviewer brings up several good points.*

*To address the DOF distribution we added a Figure in the paper showing the global distribution of the average annual DOFS corresponding to the 2013-2017 global concentration map.*

[Figure]

*Considering all the points globally for the whole 2013-2017 period the pre-filtering of the data by the ammonia spectral signal removes about 35% of the data on average. This number will vary depending on location and season. Added to the text: "This cloud pre-screening also improves the data processing rate as it reduces the number of potential number of retrievals globally by an average of 35%, with this rate varying depending on location and season."*

*The ammonia retrievals do have limited vertical information. To show example plots of retrieved surface VMR values vs the integrated retrieved profile total column values is just a matter of choice. The total column values have the advantage of being a single integrated value, where the surface VMR values have the advantage of being more applicable to air quality applications. Both the surface and the total column values are highly correlated with retrieval levels in the "boundary layer". The most accurate metric from a satellite information centric perspective is more of a Representative Volume Mixing Ratio (RVMR) values (Shephard et al., 2011), which essentially provides a "boundary-layer" volume mixing ratio value that is a vertically weighted average of the retrieved profile by the shape of the averaging kernel. Since the sensitivity varies from profile-to-profile the RVMR values, as presently prescribed, will be reported at several different levels on maps, which for this more general paper we feel is too detailed. We did consider showing both the total column and surface VMR throughout the paper, but this was too repetitive as the general hotspots were very similar since ammonia is typically short-lived and thus higher agricultural concentrations are generally close to the sources. To indicate this we did provide in the appendix some total column plots to go along with surface VMR plots shown in the main text, however, we noticed that we did not add any comments to go with them. To make this point more apparent to the reader we added following the lines in Section 3. "The corresponding plot of total column values provided in Figure A 3 show similar spatial seasonal patterns seen the retrieved surface values. This is generally expected as ammonia is typically short-lived in the boundary layer so higher agricultural hotspots are close to source locations, plus both surface level retrievals and the corresponding integrated total column values are correlated with the profile retrievals in the boundary layer where the satellite typically has maximum sensitivity (as shown in Figure1)."*

Here is a map of # of unique days with observations for each grid point (0.05 x 0.05 degrees or ~5x5 km) across North America for a 5-year period (2013-2017) with quality flag = 5.

[Figure]

2. As stated by the authors, the DOFS are generally between 0.1 and 1 (rarely above 1 if I understood it well). Is it therefore relevant to present surface concentrations? Why don't you present integrated columns when you show CrIS $NH_3$ distributions?

The total errors shown in figures 3 and 4 also strongly encourage the use of the total columns. Regarding this issue, could you comment the differences that can be observed between the 5-year averaged surface concentration distribution shown in Figure 5 and the 5-year averaged total columns distribution shown in Figure 1 of Dammers et al., ACP 2019 paper (apparently made with the same dataset).

In the model comparison, why not comparing directly the total columns instead of surface concentrations? I am also curious to have a look at a global distribution made only with observations characterized by a DOFS above 0.5. Would you mind providing such a figure?
Finally, when calculating the estimated emissions based on CrIS NH3 measurements, you only keep the observations with a DOFS above 0.8. How this threshold has been defined? Why not always using this threshold? How does it affect your results?

*As noted in response to question #1, there are pros and cons to presenting the retrievals in total columns and surface level VMR, with both be highly correlated with the upper boundary level retrieved values as there is limited independent vertical information at the surface. Even the integrated absolute total column amounts values are dominated by near surface contributions, as this is where the most molecules are located.*

*The errors in Figure 3 and 4 are random errors for single pixel. Therefore, any single retrieved value will have a significant random error, with a single level having more than an integrated profile total column value. However, most of the results shown in this paper use mean results and since random errors reduce approximately as the square root of the number of points it will not make any difference from an error point-of-view if shown the plots are shown as total column or surface volume mixing ratio values.*

*For the purpose of this emission study the 0.8 DOFS threshold was defined to as a balance between keeping points with maximum information and having enough points.  In principle you will have a slightly higher bias, but as all observations will in also have that bias these should be accounted for in the background parameter in the emissions estimate (Dammers et al., 2019).*

3. I was surprised to note that the authors do not refer to relevant work made on the remote sensing of atmospheric NH₃ using the IASI satellite, such as the first global distribution provided from space measurements presented in Clarisse et al., NatGeo 2009 (doi: 10.1038/ngeo551) and the first estimation of NH₃ emissions from industrial and agricultural sources in Van Damme et al., Nature 2018 (doi: 10.1038/s41586-018-0747-1). At the very least, these two papers should be cited.

*We did make an error as the idea when referring to each instrument that measures tropospheric NH3 is that we use the first significant NH3 paper.  Thus, for IASI this would be Clarisse et al., 2009., so the Clarisse et. al., 2010 was changed to Clarisse et al., 2009.  In addition, for the other main sections, the Van Damme et al., (2018) reference was added in the emissions section, the van der Graff et al., (2018) was added in the dry deposition section, and the Van Damme et al., (2014) was added in the model evaluation section of the paper.*

Minor comments:

· Several papers are not referenced in the bibliography:
Divarkarla et al., 2014 ; Li et al., 2019 ; Ellis et al., 2013 ; Losdale et al., 2019

*Thanks, added the 4 references into the paper.*

· Units are generally missing in the captions of the figure, I would add them

*The units are provided in the plot labels themselves, which should be sufficient for the reader.*

· Consistency in the writing of "hotspot" (currently, we have in the paper: hotspots (e.g., p1), "hotspots" (e.g., p8), hot spots (e.g., p9), hot-spot (e.g., p17) and "hot-spot" (e.g., p17))

*Good catch; they have now all been changed to hotspot throughout the text.*

· P2, L26: change Clarisse et al., 2010 to Clarisse et al., 2009

*Yes, as stated above the idea here was to reference the first NH3 paper for each instrument so the reference here was changed to Clarisse et al, 2009.*

· P3, L15-18: What is impact of the change in Level 2? From what can be seen in Figure 9, it looks not negligible for some regions. Could you comment on this in the text?

*This is a good question. The Level 2 inputs from both sources are obtained using CrIS retrievals so one would expect the impact to be minimal. However, to provide a quantitative answer we would need to perform the retrievals using both sets of level2 inputs where there is a small overlap between the products. This is a good future exercise, especially if trend analysis is to be performed. Presently we just want to indicate that there was a change by showing it in Figure 9.*

- P3, L26-30: Have you changed the pre-filtering following that update? It should be specified in the text.

*Only performing retrievals where there is an ammonia spectral signal has always been applied to the CrIS retrievals, so no updates. There is no need to perform retrievals where there is no signal, which is due to either cloudy conditions or the scene being below the detection limit of the sensor. As noted in the conclusions, a related future refinement is to use VIIRS to identify the cloudy pixels, and for the cloud-free pixels account for these non-detects in composite products.*

- P14, Figure 8: this figure could be moved to the supplementary materials (or at least reduced in size)

*We agree that the size of the figure can be small in the final manuscript.*

- P15-16, Figure 10: What is striking in this figure is the increase in 2017. While this is mention later in the text, the description of that figure does not mention it.

*Yes, this is a good point. The increase in 2017 is very evident. 2017 was a dry and hot year and a detailed more quantitative analysis required to answer this question is out of the scope of this overview paper. However, from our CrIS surface validation efforts with AMoN surface network (Kharol et al., (2020) in preparation) there is supporting evidence of the 2017 maximum. This is shown in the plots below. We added to the text the following: "Most regions over the U.S. show a maximum in surface concentrations during this 5-year period occurring in 2017, which is consistent with corresponding surface AMoN stations across the U.S. (Kharol et al., 2020). "*

[Figure]

[Figure]

· P17-18, Figure 11: Lethbridge is highlighted as a place where the model match well the satellite data. However, if you look at the southern part of Alberta, this is not really the case. Could you comment on this? While I agree that overall, there is a good match between the model and the satellite distributions, I miss some discussion on the large regions where the model disagree with CrIS. Eastern US is an important example to discuss but also Mexico (while introduced when listing the emission inventories, it is not discussed at all in the text).

*Here we were just trying to say "In general" there are elevated ammonia over Lethbridge region in both the model and satellite.  We agree that there are still differences (as shown in the emissions section) and the wording "match well" is too strong for the message.  We changed "match well" to "spatially co-located". Yes, there are differences in the absolute magnitude of some of the hotspots.  In the middle to eastern US the model tends to have larger elevated values in the more Northeastern part, where the satellite tends to show larger values then the model in the U.S. Midwest. Comparing the model emission and concentration fields in Figure 11 would indicate that these differences are mainly due to the input emission fields.  This paragraph was modified to:*

*"In general, the locations of the elevated ammonia "hot-spot" regions in simulated model surface concentration map in Figure 11b match wellspatially co-located with those observed by CrIS (Figure 11c).  This is seen in the hot-spot regions such as Lethbridge, AB and southwestern Ontario in Canada, the Central Valley in California, Washington State, Idaho, the Midwest (e.g. Nebraska) and North Carolina in the U.S.  The peak values in the hotspots in the upper Midwest and eastern U.S. for these two months in 2016 are generally higher in the model than the satellite observations., The satellite tends to show higher values than the model in the Central US (e.g. Nebraska). Comparing the model emission and concentration fields in Figure 11 would indicate that these differences are mainly due to the input emission fields. whereas in the western part of the U.S. the satellite observations tend to be*

*slightly higher over regions with elevated NH3, most notably over the Central Valley and High Plains. Similar results (not shown here) are also seen with other chemical transport models (e.g. GEOS-Chem) using the same U.S. EPA emissions inventory. whereas in the In addition, the western part of the U.S. the satellite observations tend to be slightly higher over regions with elevated NH3, most notably over the Central Valley and High Plains. There are also several elevated regions in the satellite observations in Mexico that appear to be underreported in the emission inventories."*

- P23, Figure 15: Having in mind the data filtering ("cloud screening", DOFS>0.8, only warm season considered) and the different adjustments applied on the calculate emissions, I found the error bars presented in the figure not really meaningful and think that it could be misleading (even if the caption lists what is included in the uncertainty estimate).

*The adjustments were made with the best available knowledge on the variability in the emissions to correct for season etc. As described, the remaining uncertainties follow from the uncertainties in the fit, the satellite total columns and the meteorology which have all been well described and tested by Dammers et al., 2019. In our opinion, it would be more misleading not to show the error bars. Furthermore, we would like to have added an error bar for the emission inventories, except for the fact that most of the inventories do not give an uncertainty for the gridcells/locations/sources, which gives the misleading idea that the inventories are highly accurate. As an example of the uncertainty, the EEA rates the quality of most NH3 emission categories with a "D" or "E", which corresponds to an uncertainty of 100% up to ~1000% for the individual emission factors (EEA, 2013).*

*Reference: European Environment Agency: EMEP/EEA Air Pollutant Emis-sion Inventory Guidebook, 2013 edition,http://www.eea.europa.eu/publications/emep-eea-guidebook-2013(last access:9 September 2013), 2013.*

Technical corrections:
- P2, L6: $NH4_+$ -> $NH_{4+}$

*Fixed as suggested.*

- P2, L18: add a comma after "(hours to a day)"

*Fixed as suggested.*

- P3, L16: Cross-Track -> Cross-track

*Fixed as suggested.*

- P3, L18: CPFR -> CFPR

*Fixed as suggested.*

- P3, L31: Since he -> Since the

*Fixed as suggested.*

- P4, L21: 1to -> 1 to

*Fixed as suggested.*

- P4, L22: elevationsin -> elevations in

*Fixed as suggested.*

- P5, L15: a bracket is not closed -> add ")" between "error" and "are"

*Fixed as suggested.*

- P9, L18: consider rephrasing "(do that there is bi-directional flow)"

*Removed from end of sentences, and added "(bi-directional flow)" earlier in sentence after "…re-released into the atmosphere later (bi-directional flow) depending on…"*

- P12, L7: norther states -> northern states ?

*Fixed as suggested.*

- P17, L23: two month summer period -> two-month summer period

*Fixed as suggested.*

- P17, L16-17: consider rephrasing: "The GEM-MACH-ready hourly gridded NH3 emissions at 10-km resolution over a North American(NA) grid […]"

*Fixed. Changed to: "The GEM-MACH-ready hourly gridded NH₃ emissions at 10-km resolution over North American (NA) were generated…"*

· P18, L7: "Munoz-Aluzar" does not correspond to what is in the reference list ("Munoz Alpizar")

*Fixed to be Alpizar.*

· P19, L6-7: Consider rephrasing "The atmospheric deposition of NH3 contributes excessive reactive nitrogen into water that contributes to eutrophication. "

*Fixed. Changed to: "Excessive atmospheric deposition of NH₃ adds reactive nitrogen into water that can contribute to eutrophication.*

· P19, L20: "50 U.S. states are mostly located in central and western U.S. show greater […]" -> "50 U.S. states (mostly located in central and western U.S.) show greater […]"

*Fixed as suggested.*

· P22, Figure 14: the caption misses the time period (and units) of the data used

*Added the time-period to the first sentence, "Plot of the CrIS-NH₃ total column observations from the 5-year (2013-2017) period over Lethbridge, Alberta, Canada. ", The units are in the figure itself.*

· P23, Figure 15: CrIS* -> CrIS

*Will remove the * for final published plot.*

· P24, Figure 16: rephrase "Plot showing the monthly emissions." E.g.: "Time-series of the monthly NH₃ emissions (kt/day) over Lethbridge (AB, Canada)."

*Modified to, "Time-series of monthly NH₃ emissions over Lethbridge (AB, Canada)."*

· P37, Figure A 3: I would saturate a bit more the difference plot -> colorbar from -1 to 1 e16 for example?

*Changed as suggested.*

- P32, L1: PM2:5 -> PM2.5

*Fixed as suggested.*

- P32, L15: retrievalfrom -> retrieval from

*Fixed as suggested.*

- P33, L4: reference "2014 NWT Fire Season Review Repot" to be revised

*Changed the reference to NWT_FSRR, 2015.*

- P38: consider rephrasing sentence L1-3

*Rephrased sentence to be: "There was a multisector survey in 2005 that targeted ammonia related practices from feed quality, housing and storage facilities, land application practices, and grazing management."*

---

## Author Comment (AC2) · 3 Jan 2020

Responses to the reviews for the manuscript titled, "Ammonia measurements from space with the Cross-track Infrared Sounder (CrIS): characteristics and applications".

We would first like to thank the Reviewers for taking the time to review the manuscript as their comments and edits strengthened the paper. Below you will find responses to each of their comments.

**Reviewer #3:**

The manuscript entitled "Ammonia measurements from space with the Cross-track Infrared Sounder (CrIS): characteristics and applications" describes the strategy of ammonia retrieval from CrIS on Suomi-NPP and availabilities of the product. This paper investigates the capabilities of the product for monitoring, model evaluation, dry deposition, and emission estimates. This work is quite important and largely contributes to the atmospheric chemistry community. I recommend publishing the paper after addressing several questions and comments.

Major comments:

1) Cloud filtering is mentioned in section 3.1.1. Why didn't you use the VIIRS data in the current system? In Fig.2, several scans over clouds are not eliminated. The cloud filtering described in the paper seems not so accurate and likely occur some bias to ammonia concentrations. The author should add some evaluations of cloud filtering. In addition, I would suggest that the description of the cloud filtering strategy is moved to section 2.

*The reviewer brings up a very important point in using VIIRS for cloud filtering. In terms of using VIIRS in the operational CrIS NH3 retrieval, we know of one prototype VIIRS product that is mapping the VIIRS footprints onto the CrIS footprint. However, presently there is just a few days available for this prototype product. Once this mapping product is available globally covering the CrIS timeframe we will use the VIIRS to identify cloudy CrIS pixels. This is noted in the text, "Algorithm refinements such as directly incorporating a newly developed coincident VIIRS cloud products mapped onto the CrIS footprints to distinguish the pixels with no ammonia signal due to cloud from those that have concentrations levels below the detection limits are presently being tested." Essentially, using VIIRS to identify if the pixel in cloudy, and the reason why there is no NH3 spectral signature.*

*The cloud filtering does perform well, especially for clouds with cloud optical depths > 1.0, which will make the underlying ammonia spectral signature opaque and filtered out. This is shown in Figure 2 and Figure 6 by comparing the cloud and CrIS ammonia maps. The conditions that can have some impact are the very optically thin clouds that are warmer near the surface. However, looking closely at edges of clouds, etc., this does not have much of an overall impact. This is stated in the text, "Note that thin clouds (cloud optical depth < 1.0) that are near the surface with cloud-top temperatures close to the surface temperature still impacts the current ammonia retrievals, but in general has a non-significant impact on the overall results as seen in the examples in Figure6". A more quantitative evaluation of the cloud filtering will done in the future at the same time as when the mapped VIIRS cloud product on the CrIS footprint is available. Essentially, comparing the current v1.5 with the future version that has the VIIRS cloud mask incorporated.*

*The cloud discussion section was moved into Section 2 as suggested.*

2) Page 18, line 2: Why did you select that two months, July and August? How is the consistency in the other seasons that there are no obvious wildfires? The figure for the difference between Fig. 11a and 11b is also helpful.

*One of the important sources of NH3 emissions is agricultural activities. We are conducting a research project to evaluate agricultural NH3 emissions during warm season using CrIS observed NH3. These two months were selected just as an example showing an application for such model evaluation. Similar consistency for comparisons among emissions, model predictions, and CrIS observations were seen for other months, however, we are still in the process of performing a detailed quantitative analysis. We did add a little more discussion to this section (see in the response to Review #1 "P17-18, Figure 11:" question).*

Minor comments:

1) Figure 1: The account for the circle and bar plots in the right figure should be added in the caption.

*Good catch. Added the line in the caption: "The box-and-whiskers showing the statistics (e.g. median, percentiles, and outliers (circles)) of the rows of the averaging kernel values at each retrieval level are also provided on the averaging kernel plot."*

2) Page 4, line 22: "elevationsin" should be "elevations in".

*Fixed as suggested.*

3) Page 5, line 15: ")" corresponding to "(includes: : :" is missing (after "(or smoothing) error"?).

*Fixed, added a ")" after error.*

4) Figure 5: Some other papers (e.g. Van Damme et al., 2014) and Warner et al. (2016)) reported the high concentration in Siberia and Alaska. I would recommend expanding the latitudinal area to around 70N. If there aren't significant high values, it is also valuable information.

*The global plots were expanded to 70N. The higher concentrations in Siberia and Alaska are likely due to large episodic forest fires that still impact a multi-year average. CrIS having more sensitivity than AIRS or IASI has the potential to contain lower retrieved values in these regions contributing to the multi-year mean over this region.*

[Figure]

5) Figure 5: Are the plots only over land? If so, it should be added to the body text or caption.

*Added to caption as suggested.*

6) Page 9, line 30: "SCCP" should be "ISCCP".

*Fixed as suggested.*

7) Page 12, line 7: "norther" should be "northern". 8) Page 17, line

*Fixed as suggested.*

8: "Gong et al., 015" should be "Gong et al., 2015". 9) Page 17, line 26: "aggrees" should be "agrees".

*Fixed as suggested.*

---

## Author Comment (AC3) · 3 Jan 2020

Responses to the reviews for the manuscript titled, "Ammonia measurements from space with the Cross-track Infrared Sounder (CrIS): characteristics and applications".

We would first like to thank the Reviewers for taking the time to review the manuscript as their comments and edits strengthened the paper. Below you will find responses to each of their comments.

**Referee #4:**

This manuscript by Shephard et al. titled "Ammonia measurements from space with the Cross-track Infrared Sounder (CrIS): characteristics and applications" is well written and very easy to follow. They describe well about the CrIS NH3 product and demonstrate the capabilities of this product for multiple applications, such as model evaluation and emissions estimates. The manuscript does a thorough job of describing the CrIS NH3 CFPR algorithm and its various components. The demonstration of the applications of this dataset is also comprehensive and convincible. Overall, I think it is appropriate for publication after minor revision. I have a few specific comments below.

1) Page 3, Line 24-25: "higher concentration : : : near the surface. This is demonstrated later in Section 3.2 with model emissions and corresponding simulated surface concentrations." I think this sentence is not appropriate. The higher concentrations in the model cannot demonstrate the findings from the CrIS since the emissions always emit at the surface level in the model and thus will of course get higher simulated surface concentrations.

*The point here is that the spatial patterns in the ammonia emissions and concentration field from the model highly correlated. Thus, regions with higher surface ammonia concentrations are likely regions with higher emissions. This is not always the case, for example longer lived atmospheric species where large concentrations can be significantly far from sources.*

2) Page 24: The paragraph of introducing the "31-day moving window" is not very clear to me. What is the purpose of developing these daily sets? Why do they choose 31 days? Also, I'm not sure how it helped "better capture the warm season timing of emissions".

*We chose a set of 31 days to have enough observations for an estimate. In most Chemistry Transport Models (CTMs) the emissions are prescribed, with usually a fixed seasonal, daily and hourly variation for the emissions. The daily set is helpful to better constrain the timing of for example the spring peak, following the spreading of fertilizer on fields in spring.*
*Added to the text: "This type of analysis can be used to help better constrain the seasonal and daily timing of emissions over this region."*

Typo:

Page 3, Line 31: Since he -> Since the.

*Fixed as suggested.*

Page 5, Line 6: Figure A 1 -> Figure

A.1.

*To be consistent, for the figures in the appendix changed to removing a space (e.g. Figure A1) and made same in text.*

---

## Author Response (AR1)

January 3, 2020

Dear Editor,

We provided detail responses all three reviewers' comments and questions for the manuscript titled, "Ammonia measurements from space with the Cross-track Infrared Sounder (CrIS): characteristics and applications". The manuscript was also updated based on their suggestions. We would like to thank the reviewers for taking the time to review the manuscript as their comments certainly strengthened the paper. I also have a track-changes version that can be provided if they would also like to see the changes from the previously submitted version.

Thanks,

Mark Shephard

Environment and Climate Change Canada
4905 Dufferin Street, Toronto ON M3H 5T4
Telephone : 416-739-4437
Email : Mark.Shephard@canada.ca

[revised manuscript text omitted]

**CrIS NH$_3$ Total Column (2013-2017)**

(DJF) (MAM)

(JJA) (SON)

[x 1e16 molecules cm$^{-2}$]

Figure A.32. Similar to Figure 8Figure 7 but for total column amounts.

[Figure]

[Figure]

Figure A 43. Similar to Figure 11Figure 10 but for total column ammonia amounts.

.

**Appendix D:  Brief description of Canada's Air Pollutant Emission Inventory (APEI)**

The AEAI inventory calculation (from agricultural sources) is a multi-stage process involving three sets of information. The inventory is built on very detailed census data on animals for each census district collected annually by Statistics Canada. The data on fertilizer use, including forms of nitrogen, is provided by the fertilizer industry on a provincial basis. Ammonia emissions are strongly influenced by farming practices such as manure handing systems or fertilizer

10    application method. The practices data were acquired by farm surveys, targeted to ammonia with an emphasis on timing of

practices, across the main livestock sectors across the 12 key ecoregions across Canada. There was a multisector survey in 2005 that, targeting ammonia related practices from feed quality, to housing and storage facilities, to land application practices, and grazing management. The important beef sector survey was updated in 2011 to capture large changes in practices. A new pig survey will be conducted this year. A separate survey for fertilizer practices relating to ammonia was

5    conducted in 2006 by the polling company IPSOS.

Emission factors for the particular farm practices were obtained from scientific studies conducted in Canada and elsewhere. Some published models for emissions were used, and where possible tested with Canadian data. The emission factors were adjusted for ambient temperatures relating to the practices, the regions and the time of the practices. The manure application emissions were also adjusted for the probability of rainfall. The emission data was granulated to a 50x 50 km grid

10    by ECCC and a finer grid is being contemplated. Note that in some cases, notably where there are few operations, the data is averaged over a larger areas with more operations to ensure confidentiality for the farms.